# Spatial mapping of mobile genetic elements and their bacterial hosts in complex microbiomes

Benjamin Grodner[1], Hao Shi[1,2], Owen Farchione[1], Albert C. Vill[1], Ioannis Ntekas[1], Peter J. Diebold[1], David T. Wu [3], Chia-Yu Chen[3], David M. Kim[3], Warren R. Zipfel [1], Ilana L. Brito [1] & Iwijn De Vlaminck [1] ✉

The exchange of mobile genetic elements (MGEs) facilitates the spread of functional traits including antimicrobial resistance within bacterial communities. Tools to spatially map MGEs and identify their bacterial hosts in complex microbial communities are currently lacking, limiting our understanding of this process. Here we combined single-molecule DNA fluorescence in situ hybridization (FISH) with multiplexed ribosomal RNA-FISH to enable simultaneous visualization of both MGEs and bacterial taxa. We spatially mapped bacteriophage and antimicrobial resistance (AMR) plasmids and identified their host taxa in human oral biofilms. This revealed distinct clusters of AMR plasmids and prophage, coinciding with densely packed regions of host bacteria. Our data suggest spatial heterogeneity in bacterial taxa results in heterogeneous MGE distribution within the community, with MGE clusters resulting from horizontal gene transfer hotspots or expansion of MGE-carrying strains. Our approach can help advance the study of AMR and phage ecology in biofilms.

Understanding the complex biology of mobile genetic elements (MGEs) is crucial for manipulating microbiomes and improving the treatment of microbiome-associated diseases. MGEs carried on plasmids can confer adaptive traits, including antimicrobial resistance (AMR) and virulence, to host bacteria, while bacteriophages can drastically alter the structure of microbiomes[1–3]. The host range of MGEs varies widely—some have a broad host range, while others are restricted to a single strain or species. This host range is consequential; for example, the host range of bacteriophages can impact their utility for precision microbiome manipulation or infection treatment[4]. Similarly, the host range of AMR plasmids may inform the extent to which a microbiome can act as a reservoir for AMR traits[5,6].

Despite the centrality of MGEs in microbial ecology, basic facts about the mechanisms of the spatial spread of MGEs within natural communities remain unknown. This knowledge gap largely stems from a lack of spatially resolved tools to examine the mobile gene pool in situ and to directly establish MGE–host associations. Sequencing-based approaches for linking MGEs with their microbial hosts involve dissociation of the sample and do not retain spatial information, while building reporter constructs into MGEs is limited to tractable systems[7].

In this study, we introduce an imaging-based approach that integrates single-molecule DNA fluorescence in situ hybridization (FISH) and highly multiplexed ribosomal (r)RNA-FISH to map MGEs and their cognate bacterial hosts at the resolution of a single bacterial cell[8,9]. We show that this method allows us to study the heterogeneity in the spatial distribution of MGEs within biofilms and establish links between MGEs and their hosts in complex structured microbiomes. We developed this method for confocal microscopy with spectral detection to situate MGEs in three dimensions within dense biofilms

[1]Meinig School of Biomedical Engineering, Cornell University, Ithaca, NY, USA. [2]Kanvas Biosciences, Inc, Monmouth Junction, NJ, USA. [3]Division of Periodontology, Department of Oral Medicine, Infection, and Immunity, Harvard School of Dental Medicine, Boston, MA, USA. ✉e-mail: vlaminck@cornell.edu

and to enable simultaneous highly multiplexed identification of bacterial taxa. We first assessed and optimized single-molecule DNA-FISH techniques on the basis of in situ signal amplification to ensure sensitive and specific detection of target DNA within individual bacterial cells via confocal microscopy. Next, we developed a semi-automated image analysis pipeline to detect MGE spots and segment bacterial cells. We then applied this methodology to examine the spatial spread of AMR gene-carrying plasmids and prophage in human oral plaque biofilms. We demonstrated the ability to establish MGE–host associations, and we found that both bacterial taxa and their MGEs exhibit intricate spatial structure, forming clusters within plaque biofilms on the order of 10–100 μm. This spatial heterogeneity implies the existence of limited microscale regions of horizontal gene transfer (HGT) or clonal expansion in dense biofilms and, potentially, taxonomic and physical barriers for the spread of MGEs.

## Results

### Optimization of single-molecule MGE-FISH

We used *Escherichia coli* transformed with pJKR-H-tetR plasmids encoding an inducible *GFP* gene as a model system to assess and optimize MGE-FISH on a confocal microscope (Fig. 1a)[10]. We designed FISH probes for the non-coding strand of the *GFP* gene, used non-transformed *E. coli* as a negative control and tested six different FISH protocols. Initial attempts using single and ten encoding probes yielded little to no separation between the signal in the plasmid and control samples (Fig. 1b, rows 1 and 2). This was expected given the photon noise and losses inherent in confocal microscopy as compared with a wide-field microscope[11,12]. We next implemented two enzyme-free amplification methods to increase the signal[13,14]. Branched amplification yielded a higher true positive signal, albeit accompanied with a high background signal in the negative control (Fig. 1b, row 3). Hybridization chain reaction (HCR) similarly enhanced the signal at the expense of a high background in the control (Fig. 1b, row 4). To improve specificity, we adopted a 'split' HCR method and used heat-denatured DNA and non-fluorescent 'helper probes' to stabilize the DNA[15,16]. This resulted in a significant reduction of the signal in the negative control (Fig. 1b, row 5). Last, to address autofluorescence in oral biofilms (as detailed below), we applied a gel embedding and clearing technique, in which nucleic acids in the sample are covalently anchored to a polyacrylamide gel, followed by clearing of proteins and lipids[17,18]. This method led to a high specificity of MGE detection (false positive rate <0.01) but a relatively low sensitivity (true positive rate = 0.39). We suggest that this limited sensitivity is a result of tight packing of the transcriptionally repressed *GFP* gene, limiting accessibility, as detailed previously and as supported by our experiments with a phage infection model described below[19–21]. We applied the final optimized method in conjunction with super-resolution airyscan imaging to examine the subcellular localization of plasmid-encoded *GFP* in *E. coli* cells. We found that the plasmid density is ~50% higher on average at the poles compared with the centre (Extended Data Fig. 1a,b), in line with previous reports that plasmids have limited capacity to diffuse through the nucleoid at the cell centre and tend to cluster at cell poles[22,23].

## Visualizing phage infection

Building on the optimized MGE-FISH method (Fig. 1b, row 6), we turned our attention to visualizing T4 phage infection of *E. coli*. We staged infections at four multiplicities of infection (MOI 0, 0.01, 0.1 and 1) and fixed replicate cultures every 10 min over a 40-min period (Fig. 1c and Extended Data Fig. 1c). We designed FISH probes targeting the non-coding strand of the gp34 gene, which encodes a tail fibre protein, and quantified cells with 5 or more MGE spots, less than 5 spots and no spots (Fig. 1c and Extended Data Fig. 1d). For non-infected controls (MOI 0), the fraction of cells with phage detected was 0.015 (8,800 cells, 3 fields of view), which gives the false positive rate. No cells in the MOI 0 control had more than 5 spots, which gave us confidence that the striking signal from cells with high spot count in MOI 0.01, 0.1 and 1 was specific to phage infection. We predicted the fraction of infected cells to be 0, 0.01, 0.10 and 0.73, for MOI 0, 0.01, 0.1 and 1, respectively (Poisson probability mass function). This was close to the observed fraction of cells with phage spots at 20 min: 0, 0.02, 0.23 and 0.53. This indicates much higher sensitivity than what we observed in the GFP plasmid experiment (Fig. 1b, row 6). We suggest that the actively replicating *gp34* gene is more accessible to FISH probes than the transformed, unexpressed *GFP* gene in the plasmid experiment.

T4 phage infecting *E. coli* in LB medium has a reported average latent period lasting 18 min, end of lysis at 36 min and a burst count of 110 (ref. 24). We observed MGE-FISH spots within 10 min of phage introduction, which indicates that we are visualizing replicated phage genetic material before disruption of the cell membrane. At 20 min, cells with high phage count were often physically longer in length than uninfected cells, suggesting bacterial growth with stalled division near the end of the latent period. Our results match previous findings that burst sizes for T4 phage increase with increased bacterial growth rate due to large cell volumes delaying full lysis[24,25]. We observed a dramatic increase in the fraction of infected cells for MOI 0.01 and 0.1 at 40 min. This corresponds to the expected lysis time and the adsorption of new phage to uninfected cells. At 30 and 40 min, many cells with a high phage count had a low 16S rRNA signal and increased width and length compared with uninfected cells (Fig. 1c and Extended Data Fig. 1c). We suggest that these cells with high phage count and low 16S rRNA intensity have been fully lysed, meaning that MGE-FISH can be used to stain encapsulated phage particles, as has been suggested previously[26]. We also observed a small fraction of infected cells with a low 16S rRNA signal in the centre of the cell and a high signal at the poles (Extended Data Fig. 1d, middle), which we suggest are infected cells that experience cytoplasmic condensation due to membrane damage[27]. Overall, these data and observations match the expected progression of a T4 phage infection course and show the value of MGE-FISH imaging in generating insights even in a well-studied system.

### Mapping MGEs in oral plaque biofilms at high specificity

Next, we evaluated the ability of our MGE-FISH method to visualize the spatial distribution of MGEs in human oral plaque biofilms. To this end, we collected oral plaque biofilms from two healthy volunteers (A and B) and performed shotgun metagenomic sequencing on a portion of each sample, reserving the rest for imaging (Fig. 2a). As an initial controlled

**Fig. 1 | Single-molecule MGE-FISH. a**, Diagram of *E. coli* model *GFP* plasmid system used to optimize single-molecule FISH. **b**, (**i**) Diagrams of different methods implemented. Blue cells on the left are wild type and orange cells on the right are transformed with the plasmid. After the first row, two encoding probes are shown to represent ten encoding probes in all cases. Magenta lines represent the plasmid, cyan represents 16S rRNA and blue represents off-target binding sites. (**ii**) Representative images for each method alteration. Magenta indicates a signal from MGE-FISH and cyan indicates a signal from 16S rRNA-FISH. Scale bar, 5 μm. Images were captured for at least 1,000 cells in each condition. (**iii**) Fraction of cells with spots for control and plasmid images as a function of signal-to-noise ratio (SNR) threshold. SNR was calculated for each spot, dividing the spot signal by the average background signal ('Manual spot background filtering' in Methods). Black vertical line indicates the selected SNR threshold. TPR, true positive rate; FPR, false positive rate (at the threshold). (**iv**) Histograms for the number of spots in each cell. Width indicates the frequency of the spot count value. Horizontal red bars indicate mean spot count. **c**, Left: diagram of MGE-FISH staining of *E. coli* infected by T4 phage. Middle: example images for four multiplicities of infection at 20 min and 30 min after introducing phage to the culture. Right: results of manual counting to classify cells into groups on the basis of the number of MGE-FISH spots.

test of the method (Fig. 1b, row 6), we stained for the *GFP* gene in samples that contained mixtures of plaque and *GFP*-transformed *E. coli* (Fig. 2b) and demonstrated that the specificity remained high in plaque.

Via metagenomic analysis, we identified *mefE*, an AMR gene encoding an antibiotic efflux pump, in the plaque of volunteer A but not volunteer B (Fig. 2c and Supplementary Tables 5 and 10). Our MGE-FISH

method confirmed the prediction from metagenomic analysis; we measured 0.012 and <0.001 *mefE* spots per cell in volunteers A and B, respectively (Fig. 2c and Extended Data Fig. 2a). Furthermore, we demonstrated that there was positive spatial autocorrelation of *mefE* spots in volunteer A (Moran's $I$ = 0.015, $P$ = 0.005; Extended Data Fig. 2b), suggesting that the process underlying the distribution of plasmids was

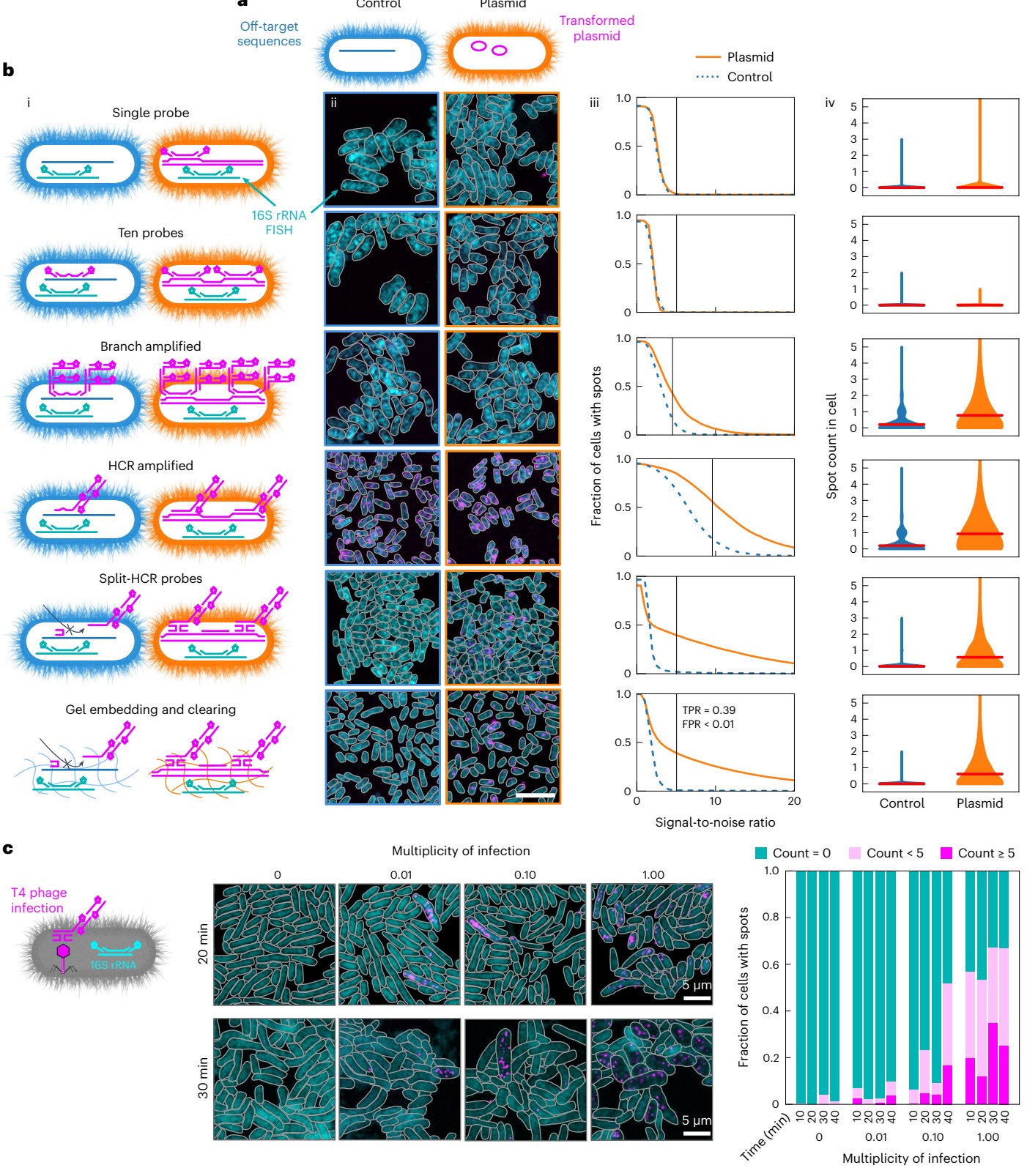

non-random, while the spots in volunteer B were randomly distributed (Moran's $I$ = 0, $P$ = 0.259). These results showed that MGE-FISH is effective in visualizing MGEs in plaque. The spatial clustering of this AMR plasmid within the biofilm suggests that we are probably observing plasmid proliferation either through host replication (vertical transfer) or conjugation (horizontal transfer)[28].

In the plaque, we observed off-target signals as bright patches and dispersed large spots, probably due to non-specific binding of probes to food particles or debris. To mitigate this issue, we implemented gel embedding and clearing for reduced off-target binding[17,18,29]. To test the efficacy of gel embedding and clearing, we used orthogonal FISH probes, designed to not target any sequence in the plaque. We observed a dramatic reduction in off-target signal after gel embedding and clearing (Extended Data Fig. 2c,d) and therefore used this in all subsequent experiments on plaque.

We next mapped a natural lysogenic bacteriophage (prophage) in plaque to study its spatial distribution. In volunteer B, we identified a T7-like prophage via metagenomic analysis and developed probes targeting its *capsB* gene, which encodes the minor capsid protein (Supplementary Tables 6 and 10). In these experiments, we used two negative controls to assess off-target binding: one with no probe and one with orthogonal probes. Both controls displayed minimal off-target signal (Extended Data Fig. 3a), and we could set an area threshold on spots to further filter out off-target signals on the basis of the spot size. *CapsB* spots clustered spatially, coinciding with long rod-shaped bacteria. The spatial clustering of this phage is probably due to a limited host range; in the metagenomic analysis this prophage was binned with *Corynebacterium*, a long rod-shaped bacterium that forms spatial clusters[30]. Large clusters (~100 µm) of host bacteria may result in prophage hotspots in a biofilm (Extended Data Fig. 3b).

To further test the robustness of MGE-FISH in plaque, we then proceeded to label another phage gene in three different colours simultaneously. We identified a highly prevalent prophage of the class *Caudoviricetes* with a large terminase-encoding gene, *termL*, and were able to design a large set of FISH probes (Supplementary Tables 7 and 10). We divided the probes into three groups, each labelled with a different colour. We mapped the large-scale distribution (~25 µm) of spots in each colour and found that they formed similar patterns, as expected (Fig. 2d). We also demonstrated that different colour spots co-localized with each other at the micron scale. Similar to the previous prophage, this prophage also formed isolated spatial clusters, suggesting spatial restriction of host bacteria within plaque biofilms. While dense clusters of host cells could result in rapid transfer of a lytic phage within the cluster, the spatial isolation of different host clusters may limit the global spread of infection, with the intervening non-host cells acting as a barrier to phage transfer.

In addition to MGEs, we also tested the possibility to visualize genes located on bacterial genomes. Using metagenomic analysis, we identified three non-plasmid AMR genes (Supplementary Tables 8 and 10). Genes *patA* and *patB*, which encode subunits of an antibiotic efflux pump, were from the same metagenome-assembled genome (MAG) and had nearly identical coverage values, so we expected them to spatially co-localize. We found another gene encoding an antibiotic efflux pump subunit, *adeF*, in a different MAG (Fig. 2e). At the large scale (~25 µm), *patA* and *patB* had similar density patterns, while *adeF* had a distinct pattern, as expected. At the micron scale, MGE-FISH staining for these three genes showed that 32% of *patB* spots co-localized with *patA*, while only 5% of *patB* spots co-localized with *adeF*. The difference in large-scale spatial distribution between *patA*/*patB* and *adeF* indicates that bacteria carrying these AMR genes inhabit different spatial structures within the biofilm. Identifying spatial patterns for AMR genes within biofilms via MGE-FISH can help gain understanding of the maintenance and spread of AMR.

## Combined taxonomic mapping and MGE mapping

We next overlaid MGE biofilm maps with taxonomic identity maps to associate MGEs with their host taxa. To start, we measured the taxonomic association of a highly abundant prophage of class *Caudoviricetes*, for which the metagenomic data and RefSeq alignment hinted at a strong taxonomic association with *Veillonella* (Fig. 3a and Supplementary Table 10)[31–34]. We used rRNA-FISH to stain five common oral genera, *Veillonella*, *Streptococcus*, *Corynebacterium*, *Lautropia* and *Neisseria*, each with a different fluorophore, and we used MGE-FISH to stain the *termL* gene of the active prophage with a sixth fluorophore (Fig. 3b). The *termL* gene and *Veillonella* showed striking co-localization, mirroring the prediction from metagenomic assembly (Fig. 3c). We quantified the fraction of *termL* spots that were nearest neighbours with each species and compared the observed values to simulations of randomly distributed spots. *Veillonella* displayed by a large margin the highest spatial association considerably above random ($z$-score = 7.7, $P$ ≤ 0.01; Fig. 3d). The fraction of *termL* spots associated with *Veillonella* was 0.39, while the fraction of *termL* spots associated with each other genus was very low (~0.01). These results demonstrated our ability to determine MGE host taxonomy in plaque biofilms by concurrently mapping taxa identity and MGEs. In this biofilm, we found cells classified as *Veillonella* co-localized with *termL* signal with unexpected filamentous morphology. These cells are stained with the fluorescent barcode we assigned to *Veillonella* and display large area patches of *termL* signal. While it is possible that these filamentous cells are not *Veillonella* and thus both the *Veillonella* 16S rRNA-FISH probes and the *termL* probes bind off-target, we suggest it is also possible that this large area *termL* signal reveals active phage replication and that the long filamentous morphology is a stress response of *Veillonella* to infection as we have observed in *E. coli* (for example, Fig. 1c, 20 min MOI 0.01 and 0.1) and others have demonstrated with other stressors[35].

Next, we sought to confirm the host of a highly abundant plasmid discovered in the metagenomic data. We assembled contigs using combined long- and short-read sequencing and identified a highly abundant plasmid. Alignment of this contig to the plasmid database (PLSDB v.2023_11_23) showed that the plasmid had previously been observed in *Prevotella nigrescens* (Fig. 3e)[36]. We selected two genes from the contig that encode proteins with metallo-β-lactamase (MBL)

**Fig. 2 | MGE-FISH in human oral plaque. a**, Diagram of the workflow to apply MGE-FISH in oral plaque biofilms (created with BioRender.com). **b**, Left: example images of plaque, transformed *E. coli* expressing *GFP*, and the combination of both plaque and *E. coli*. All samples were stained for the *GFP* gene using MGE-FISH. The experiment was repeated three times with similar results. Right: association of MGE-FISH signal with *GFP* cells and non-*GFP* cells in each sample. **c**, Left: diagram of two-volunteer control experiment. Middle: example images of plaque samples from each volunteer stained for the *mefE* gene. At least three tiled fields of view (FOVs) were collected for each sample with similar results. Right: measurement of relative spot count for each volunteer. Spot counts for each image were normalized by dividing the number of segmented spots by the number of segmented cells ('Semi-automated image segmentation' in Methods). **d**, Top left: diagram showing the multicolour approach used to stain the gene *termL*. Bottom left: example FOV plotted as separate density maps for each colour of *termL* probes. At least three tiled FOVs were collected for each sample with similar results. Inset 1: zoomed region of the plaque overlaid with all colours of *termL* stain. Inset 2: zoomed region of plaque split into each colour of *termL* probes. Right: measurements of *termL* colour co-localization normalized as the fraction of total spots. **e**, Top left: diagram showing the multicolour approach used to simultaneously stain the genes *patA*, *patB* and *adeF*. Bottom left: example FOV plotted as separate density maps for each gene. At least three tiled FOVs were collected for each sample with similar results. Inset 1: zoomed region of the plaque overlaid with all colours. Inset 2: zoomed region of plaque split by gene. Right: measurement of co-localization of *patB* spots with each other gene normalized as the fraction of *patB* spots co-localized.

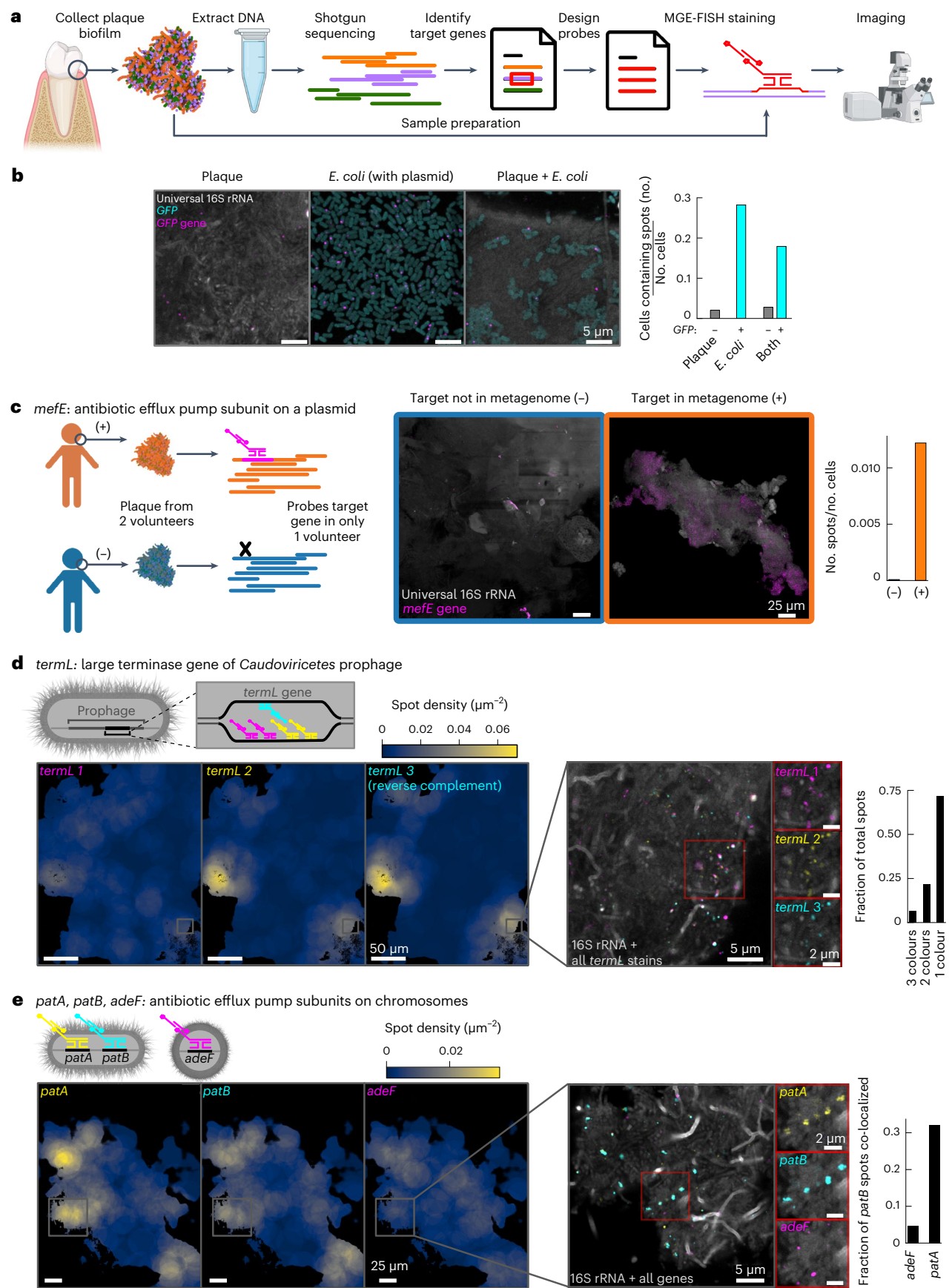

domains as targets for MGE-FISH (Supplementary Tables 10 and 11)[37,38]. *Prevotella* species are commonly resistant to β-lactam antibiotics, but most encode serine β-lactamases such as *CfxA* and are therefore generally still susceptible to carbapenems[39,40]. However, MBLs can hydrolyse carbapenems and can confer broad-range antibiotic resistance. We stained both putative MBL genes (*pMBL*) with the same colour using MGE-FISH and found dense clusters of *pMBL* at a scale of 10 μm, with these clusters commonly spaced 20–30 μm apart (Extended Data Fig. 4). For taxonomic mapping, we broadened our target panel by employing high-phylogenetic-resolution fluorescence in situ hybridization (HiPR-FISH), a method that uses combinatorial spectral barcoding to map taxa. We selected a target panel of 18 genera that are highly abundant and prevalent in human plaque[30]. We designed a HiPR-FISH spectral encoding using a 5-fluorophore combinatorial barcoding scheme, whereby each fluorophore represents a binary bit, providing 31 possible barcodes ($2^5 - 1 = 31$)[8]. The fluorophore for MGE-FISH was spectrally distinct from those of HiPR-FISH, enabling simultaneous implementation of both methods (Fig. 3f).

Using integrated HiPR-FISH and MGE mapping, we observed spatial association of *pMBL* with *Prevotella* as predicted ($z$-score = 4.6, $P$ < 0.01; Fig. 3g,h). Further, 56% of *pMBL* spots were nearest neighbours with a *Prevotella* cell. The visually and quantitatively prominent association of *pMBL* with *Prevotella* suggests that *Prevotella* is the host for the *pMBL* plasmid. In addition, we observed association between *pMBL* and *Streptococcus* ($z$-score = 11.1, $P$ < 0.01). While only 20% of spots associated with *Streptococcus*, 13% of all *Streptococcus* cells associated with *pMBL* spots. This association of *Streptococcus* with *pMBL* could be an artefact of *Streptococcus* co-localizing with the plasmid carrying *Prevotella*. It is also possible that this measurement reveals HGT between *Prevotella* and *Streptococcus*. All in all, these experiments constitute a demonstration of the use of DNA-FISH and rRNA-FISH to measure associations between host cells and MGEs and to uncover the spatial context of MGEs in dense biofilms.

Next, we investigated the taxonomic association of an unknown plasmid within a plaque biofilm of a patient diagnosed with stage 3 periodontitis. We combined long- and short-read sequencing to identify a complete plasmid with minimal homology to any sequence in the RefSeq database (v.220; Fig. 4a,b and Supplementary Table 10)[41]. The plasmid carried several predicted genes for mobilization and toxin–antitoxin systems. In the short-read data, there was a region with a dip in coverage, to zero in some bases, which we attribute to stretches of G and C homopolymers, which are disfavoured by Illumina sequencers (Extended Data Fig. 5). However, we achieved complete coverage of the plasmid with the nanopore sequencing data, which allowed assembly of the full plasmid sequence (Fig. 4b). We designed MGE-FISH probes for the plasmid and combined this MGE-FISH stain with an 18-genera HiPR-FISH panel (Supplementary Table 12). We found that the plasmid was spatially associated with *Streptococcus*

($z$-score = 26.7, $P$ < 0.01) and that *Streptococcus* formed small clusters within large patches of non-biofilm material (Fig. 4c,d). This material included host tissue, calculus and blood. We found that samples from the periodontitis patient contained considerably more of this kind of material than the healthy plaque samples. We measured 44% of plasmid spots associated with *Streptococcus*. We suggest that this plasmid is hosted by a *Streptococcus* species that is successful in the periodontitis environment.

## Discussion

Here we introduced a method for mapping MGEs in bacterial biofilms at the resolution of single cells. We optimized this method by systematically evaluating single-molecule FISH techniques to increase signal-to-noise ratio and reduce off-target binding. The resulting high sensitivity and high specificity method allowed us to map MGEs in vitro and in human oral plaque biofilm samples using confocal microscopy. In addition, we integrated our method with HiPR-FISH, a technique we previously created for bacterial taxon mapping in biofilms, allowing us to directly associate MGEs with their host bacteria and reveal correlations between local community structure and MGE spatial distribution. This versatile pipeline will be a valuable tool to generate and evaluate questions in microbial ecology.

Using this method, we were able to make unique observations about MGE distributions across spatial scales in model bacteria and human oral plaque biofilms. At the subcellular level, in vitro, we found that high-copy plasmids without partition systems show fewer puncta than expected and localize to the poles of the cells, which supports the idea that these plasmids bunch together within the cell and do not diffuse readily in the nucleoid. We also showed that there are dramatic changes in cell shape and ribosome density associated with the number of copies of a replicating phage in *E. coli*, providing unexpected insight into the physical response of cells to infection. At the 10–100 μm scale in plaque biofilms, we demonstrated that AMR genes on plasmids and chromosomes can form clusters. We further observed clustering of two prophages at the same scale in plaque biofilms, with clusters of host cells isolated from each other by intervening non-host cells. Spatial clusters of prophages and AMR genes result from either short-range MGE exchange in dense clusters of host cells or clonal expansion of MGE host cells, but we cannot distinguish between these two possibilities with MGE-FISH. We suggest that long-range (>100 μm) horizontal transfer of MGEs between clusters of host cells is limited by the need for MGEs to diffuse through the non-host biofilm. Although the literature reports that HGT is often higher in biofilms than in planktonic culture, we suggest that this observation is dependent on community spatial structure, with large variations in the local rate of HGT for a given MGE[28,42,43]. Most importantly, we demonstrated the ability of our imaging-based approach to link MGEs with their bacterial hosts, including in a scenario where metagenomic sequencing could

**Fig. 3 | Combined MGE and taxonomic mapping. a**, Workflow for prophage host association predictions via metagenomic sequencing assembly, binning and phage gene prediction. **b**, Diagram showing simultaneous single-colour rRNA stain for taxon mapping and HCR staining for prophage mapping. **c**, Top: bacterial genera classified by rRNA-FISH overlaid with the raw signal from MGE-FISH on *termL* prophage gene. Bottom left: zoomed region of rRNA-FISH overlaid with MGE-FISH. Bottom right: zoomed region showing only *Veillonella* (blue) and *termL* (magenta and yellow) in colour, while all other cells are greyscale. The arrows indicate examples of *termL* signal co-localized with *Veillonella* in magenta and *termL* signal co-localized with another genus in yellow. **d**, Left: z-scores for the number of associations between *termL* and each genus (circles) compared to simulation where the *termL* spots are randomly assigned to cells (boxplots, 1,000 simulations). The bounds of the boxes show the first quartile to the third quartile, the centre shows the median and the whiskers show the farthest data point lying within 1.5× the IQR. Right: fraction of *termL* spots associated with each taxon. Association between a cell and a spot is defined as the nearest neighbour cell to

the spot. **e**, Workflow for plasmid host association prediction via metagenomic sequencing assembly, plasmid prediction and alignment to a reference database. **f**, Diagram showing simultaneous multicolour HiPR-FISH rRNA staining for taxon mapping and HCR staining for plasmid mapping. **g**, Top: bacterial genera classified by HiPR-FISH overlaid with raw signal from MGE-FISH on *pMBL* genes. Bottom: two zoomed regions of HiPR-FISH overlayed with MGE-FISH. For all MGE-FISH spot association measurements, we filtered large non-circular signal as shown at the bottom right in *Leptotrichia* cells. **h**, Left: z-scores for the number of associations between *pMBL* and each genus (circles) compared to simulation where the pMBL spots are randomly assigned to cells (boxplots, 1,000 simulations). The bounds of the boxes show the first quartile to the third quartile, the centre shows the median and the whiskers show the farthest data point lying within 1.5× the IQR. Right: fraction of *pMBL* spots associated with each taxon. Association between a cell and a spot is defined as the nearest neighbour cell to the spot.

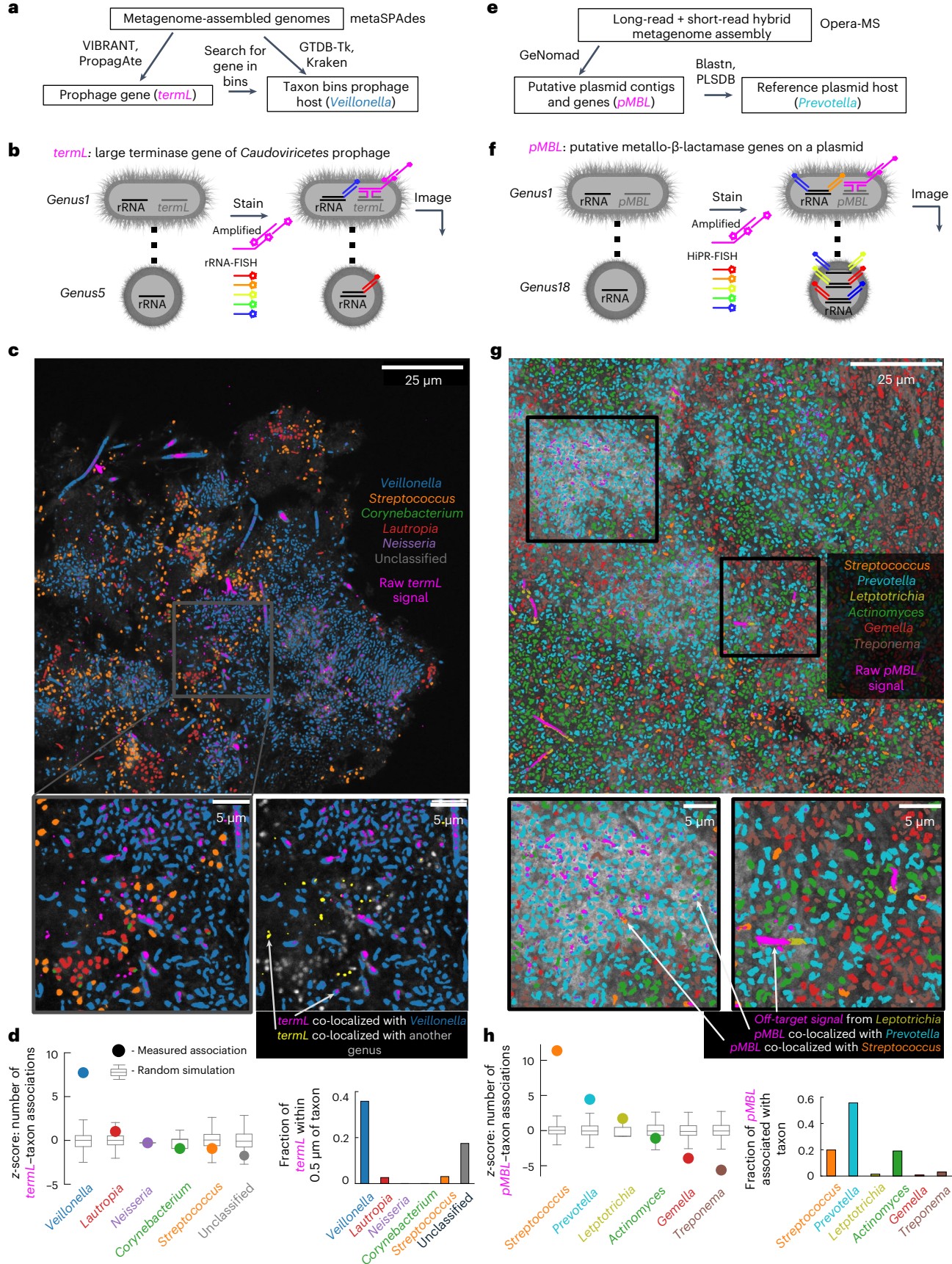

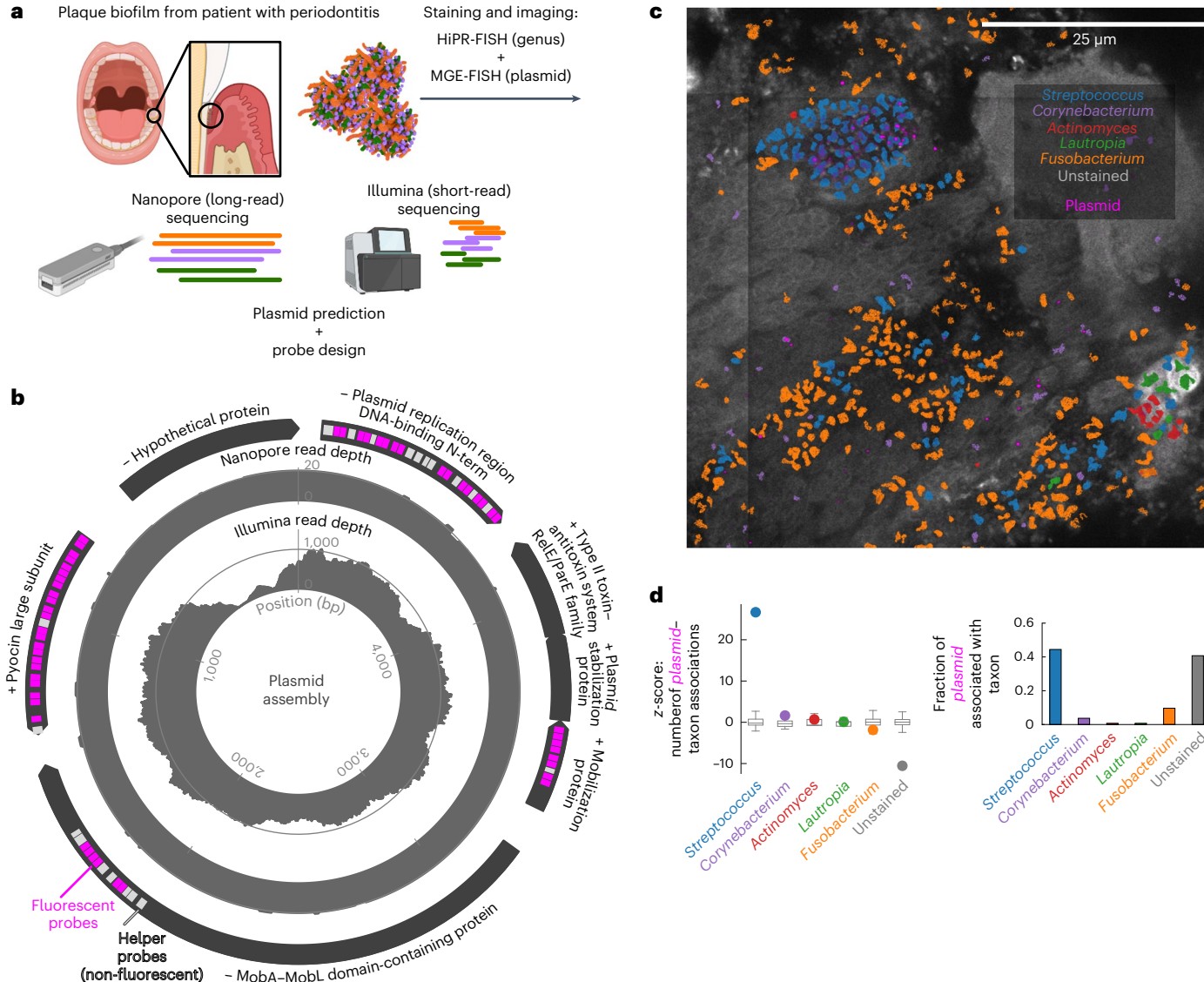

**Fig. 4 | Identifying the host taxon of a previously undescribed plasmid.**
**a**, Diagram illustrating sample collection from a patient with stage 3 periodontitis followed by DNA extraction for long-read Nanopore sequencing and short-read Illumina sequencing (created with BioRender.com). The same sample was then used for simultaneous HiPR-FISH and MGE-FISH staining. **b**, Diagram of a plasmid assembled from long- and short-read sequencing. The inner axes plot Illumina short-read alignment as reads per base and the middle axes do the same for Nanopore long reads. The outer bars plot the locations and names of predicted genes, where arrows and '+/−' at the start of the names indicate gene orientation. Pink bars indicate the locations of encoding probes for HCR staining, while light grey bars indicate the locations of non-fluorescently stained helper probes. **c**, Bacterial genera classified by HiPR-FISH overlaid with the raw signal from MGE-FISH on the previously undescribed plasmid. Grey indicates the raw signal from autofluorescence in unstained sample material. **d**, Left: z-scores for the number of associations between the plasmid and each genus (circles) compared to simulation of random distributions of the same spots (boxplots, 1,000 simulations). The bounds of the boxes show the first quartile to the third quartile, the centre shows the median and the whiskers show the farthest data point lying within 1.5× the IQR. Right: fraction of plasmid spots associated with each genus.

not. Our method provides the means to study the impact of taxonomic heterogeneity on the dissemination of MGEs in highly diverse natural biofilms.

There are remaining limitations to the method presented here. First, there is a limit to the taxonomic resolution achievable by 16S rRNA-FISH due to low rRNA sequence divergence at the species and strain level. In addition, the method relies on the availability of a microscope with both spectral and Airyscan detectors. Further, we have applied this method so far only to oral plaque biofilms, but we believe it could be transferred to other microbial ecosystems such as the gut and skin, after sample type-specific optimization. Last, the discovery of MGE targets depends on initial DNA sequencing. Complete assembly of plasmids was difficult even with high depth of coverage

from short-read sequencing but was greatly improved with long-read nanopore sequencing.

Several studies have recently demonstrated mapping of microbial RNAs using slide capture or imaging-based spatial transcriptomics methods[44–47]. Compared with MGE-FISH, these approaches offer the advantage of mapping a broader range of bacterial transcripts, but slide capture methods have lower spatial resolution and imaging-based methods have only been applied in single-species systems. Further, genes acquired from HGT are often transcriptionally repressed and therefore not detectable with transcriptomic methods[48].

We suggest that MGE mapping can serve as a direct complement for metagenomic sequencing of spatially structured microbiomes. We envision two potential application areas. First, the methods we

describe could be employed to investigate the processes that govern the emergence of antibiotic resistance. Horizontal gene transfer is considered the key mechanism by which pathogens acquire antibiotic resistance, yet fundamental aspects of MGE ecology remain unknown such as the relationship between the local physical environment and the extent of MGE transfer[28,49]. MGE mapping data could reveal physical parameters that influence HGT, such as spatial structures or spatially clustered bacterial consortia that promote or prevent the spread of resistance elements in microbiomes. Second, MGE mapping can help address the challenge of determining bacteriophage host taxa, which is crucial given renewed interest in phage therapy as an antibiotic alternative[4]. In this context, MGE mapping can further be used to examine the spatial interplay between bacteria and phages in complex ecosystems, revealing the effect of local and macro structures in biofilms on phage spread, taxonomic barriers to phage infection, varying propagation modes through biofilms, the contribution of phage to biofilm structure and biofilm 'refugia' areas with reduced phage infectivity[50,51]. These findings can then serve as a platform for developing and assessing phage therapies.

## Methods

### Inclusion and ethics statement
The protocol for volunteer recruitment and sample collection was approved by the Cornell Institutional Review Board (IRB; #2102010112) and the Harvard School of Dental Medicine IRB (IRB21-0662).

### Human participants sample acquisition
At Cornell University, volunteers were enrolled for specimen collection following informed consent and assent for the collection protocol as approved by the Cornell University IRB (#2102010112). Volunteers were asked to refrain from cleaning their teeth for 24 h. Volunteers then used the point of a plastic toothpick to scrape the plaque from the surface of a tooth just beneath the gumline on the front and back of the tooth. They then scraped the gaps on either side of the tooth by sliding the point of the toothpick into each gap and scraping away from the gums. After each scraping action, volunteers dipped the point of the toothpick into a 1.5 ml sample collection tube containing 0.5 ml 50% ethanol to deposit the plaque in the liquid. Samples were collected and stored at −20 °C until used.

At Harvard School of Dental Medicine, IRB approval (IRB21-0662) was obtained for collection of patient specimens in the advanced graduate periodontal department. The patient had complete baseline clinical measurements (pocket depth, recession, attachment loss, bleeding on probing and plaque index) and full-mouth x-rays (periapical radiographs and bitewings) to assess periodontal disease severity and establish a diagnosis on the basis of the 2018 classification (staging and grading). Following informed consent and assent, the clinicians collected supra- and subgingival plaque specimens as part of the clinical care procedure and stored them in 70% ethanol at −20 °C.

### *E. coli* transformation and preparation
Plasmid pJKR-H-TetR was acquired from Addgene (https://www.addgene.org/62561/) and transformed into *E. coli* str. K-12 substr. MG1655 (refs. 10,52). Transformed *E. coli* were streaked on LB agar Miller modification with 100 mg l⁻¹ ampicillin trihydrate (MP Biomedicals, 7177-48-2) and grown overnight aerobically at 37 °C. An isolated colony was picked and grown overnight aerobically at 37 °C with 200 r.p.m. shaking in 5 ml of LB medium Miller modification with 100 mg l⁻¹ ampicillin trihydrate. Overnight culture (100 µl) was subcultured in 10 ml modified LB with ampicillin and grown for 2 h aerobically at 37 °C with 200 r.p.m. shaking. The culture was then split in half and one tube received 40 µl 2 µg µl⁻¹ anhydrotetracycline (Takara, 631310) to induce *GFP* expression. Cultures were mixed with 10 ml 4% formaldehyde in PBS (pH 7.2 at 25 °C) and fixed for 90 min at room temperature. Fixed cells were pelleted (7,000 × *g*, 4 °C, 5 min), resuspended in 500 µl cold

PBS and transferred to 1.5 ml centrifuge tubes. Cells were washed by pelleting (10,000 × *g*, 4 °C, 3 min), resuspended in 500 µl cold PBS and washed again by pelleting and resuspending in 100 µl distilled water. Absolute ethanol (100 µl) was added to each tube to create fixed cell suspensions in 50% v/v ethanol, which were then stored at −20 °C until imaging. Wild-type cells were prepared in parallel, but without ampicillin in growth media and agar.

### Phage stock preparation
*E. coli* str. K-12 substr. MG1655 was grown overnight in mod. LB medium (25 g l⁻¹ Luria-Bertani broth, 300 mg l⁻¹ CaCl₂, 2 g l⁻¹ ᴅ-glucose). Overnight culture (5 ml) was subcultured in 50 ml mod. LB and grown aerobically at 37 °C with 200 r.p.m. shaking for 30 min, then 500 µl T4 lysate was added and allowed to infect for 5 h while shaking. Cells and cellular debris were removed from the lysate by centrifugation (7,000 × *g*, 4 °C, 10 min) and filtration through a 0.2 µm SUPOR syringe filter (Pall). Lysate titre was determined by serially diluting lysates in mod. LB and spotting triplicate 10 µl drops of each dilution onto lawns of *E. coli* plated on mod. LB agar (15 g l⁻¹ agar).

### Time-course infection experiment
Replicate 7 ml mod. LB aliquots were inoculated with 100 µl overnight *E. coli* culture and grown to optical density at 600 nm (OD₆₀₀) = 0.15 (-2 × 10⁷ c.f.u.s ml⁻¹ based on growth curve analysis). High-titre T4 lysate was diluted in mod. LB and added to each culture at a multiplicity of infection of 0.01, 0.1 or 1, with uninfected cultures serving as controls. Cultures were grown aerobically at 37 °C with 200 r.p.m. shaking. At the prescribed timepoints, cultures were mixed with 7 ml 4% formaldehyde in PBS (pH 7.2 at 25 °C) and fixed for 90 min at room temperature with continuous inversion. Fixed cells were pelleted (7,000 × *g*, 4 °C, 5 min), resuspended in 500 µl cold PBS and transferred to 1.5 ml centrifuge tubes. Cells were washed by pelleting (10,000 × *g*, 4 °C, 3 min), resuspended in 500 µl cold PBS and washed again by pelleting and resuspending in 100 µl distilled water. Absolute ethanol (100 µl) was added to each tube to create fixed cell suspensions in 50% v/v ethanol, which were then stored at −20 °C until imaging.

### DNA-FISH split-probe design
Probes were designed using a custom Snakemake 7.18.2 pipeline with rules written in Python 3.6.8 using numpy 1.15.4 and pandas (0.24.1)[53,54]. Target gene sequences were taken as inputs along with a reference blast database. The target was aligned to the blast database and all significant alignments were recorded for future filtering (blastn 2.13.0). All possible oligonucleotide probes were designed to be complementary to the coding strand of the target gene (that is, the same sense as the mRNA) using Primer3 (v.2.3.5)[55]. Pairs of Probes in this pool were identified as any probes aligning less than three base pairs distant from each other. These probe pairs were then blasted against the reference database using blastn from the US National Center for Biotechnology Information. On-target blast results were removed from the results using the target gene alignment IDs. Non-significant blast results were then filtered using user-defined parameters. These include maximum continuous homology (12), GC count (7) and melting temperature (46 °C). All blast results with values in these parameters that were less than the specified thresholds were removed as 'non-significant alignments'. The remaining blast results were considered 'significant' or likely to produce off-target signal. Probe pairs were removed when both probes had off-target homologies to nearby regions in the reference database. This nearness parameter is another user-defined threshold. The remaining probe pairs were then sorted with favoured probes having low levels of off-target homology. Going down the sorted list, probe pairs were then selected to tile along the gene without overlapping. Selected probes were then appended with appropriate flanking regions so that the target would be stained with the intended fluorophore (Supplementary Table 1). Two base-pair spacer nucleotides between the

flanking region and the probe were selected to minimize the off-target homology of the full-length probes in a manner similar to how probe pairs were sorted by blast results. The pool of selected probe pairs was then evaluated by searching for any off-target homologies where two probes were nearby each other. 'Helper' probes were then selected from Primer3 to tile along the gene without overlapping the existing probes. The final probes were then submitted for oligo synthesis to Integrated DNA Technologies (IDT) at a concentration of 200 µM.

### DNA-FISH single-probe design

Single probes were designed much as the split probes up to the Primer3 step. Then, instead of pairing probes, the probes were all blasted against the database and the blast results were filtered as the split probes were for 'significant' off-target homologies. Probes with any significant off-target homologies were removed and the remaining probes were tiled along the target gene to ensure no overlap. The selected probes were then paired with flanking regions for the readout stain, and two base-pair spacers were added and optimized as in the split-probe design. The resulting probes were submitted to IDT for synthesis.

### Orthogonal probe design

Probes with zero significant off-target blasts were selected from split probe pairs for different genes. For example, if the left probe from a pair targeting Gene A has zero off-target blasts, it is selected, then the right probe from a pair targeting Gene B is selected. The concept is that it is very unlikely for these probes to hybridize close enough to each other to initiate HCR fluorescence amplification. Three right probes and three left probes were selected in this manner and pooled to create an 'orthogonal' probe pool (Supplementary Table 1).

### Single-molecule FISH-transformed *E. coli* hybridization method development protocols

Six protocols were implemented. In the first three, fixed cells suspended in 50% ethanol were deposited on an Ultrastick slide (Electron Microscopy Sciences, 63734) and allowed to dry in a monolayer. Cells were covered in 10 mg ml⁻¹ lysozyme in 10 mM Tris-HCl pH 8.3, incubated at 37 °C for 1 h and washed for 2 min in 1x PBS. Cells were covered with hybridization mix containing encoding probes (2x SSC, 5x Denhardt's solution, 10% ethylene carbonate, 10% dextran sulfate, 200 nM MGE probes, 200 nM EUB338 probes; Supplementary Tables 1 and 2), incubated for 4 h at 46 °C, then washed for 15 min at 48 °C (215 mM NaCl, 20 mM Tris-HCl pH 7.5, 5 mM EDTA). Cells were then covered with a hybridization mix containing fluorescent readout probes, incubated for 2 h at room temperature and washed for 15 min at 48 °C. Incubations were performed using Frame-Seal slide chambers (Bio-Rad SLF0201) and washes were performed in coplin jars. Slides were dried with ethanol, mountant (ThermoFisher, P36982) was deposited on the slide, a glass coverslip was placed on top and the mountant cured for 24 h. In the first protocol, only one encoding probe sequence was used with standard single fluor readout probes[56]. In the second, ten encoding probes were used. In the third, branched readout probes were used[13]. In the fourth protocol, hybridization chain reaction readout probes were used (prepared as previously described)[14] at 60 nM, the hybridization mix for the readout probes was altered to omit ethylene carbonate and readout was time reduced to 1.5 h. In the fifth protocol, the 10 encoding probes were substituted for 10 pairs of split encoding probes[15]. In the fifth protocol, we also added a denaturation step after removing lysozyme from the slides. In this step, we covered the cells with 50% ethylene carbonate, incubated them at 60 °C for 90 s, then immersed the slide in a series of ice-cold 70% ethanol, 90% ethanol, then 100% ethanol for 5 min each. Here we also added 'helper' probes to the encoding probe mix, these 'helper' probes being unlabelled oligos with lower specificity than encoding probes, intended to stabilize the double-stranded DNA in its denatured conformation.

In the sixth protocol, we performed gelling and clearing. For this protocol, cells were deposited on 40 mm round coverslips (Bioptechs, 40-1313-0319) that had been cleaned with alconox, immersed in acidic wash (5 ml 37% HCl, 5 ml methanol) for 30 min, washed in ethanol, immersed in bind silane solution (9 ml ethanol, 800 µl distilled water, 100 µl Bind Silane (GE, 17-1330-01), 100 µl glacial acetic acid) for 30 min and allowed to air dry. Cells were then prepared as above through denaturation, then the cells were covered with Label-X solution (prepared as previously documented)[57], incubated for 6 h at 37 °C, washed in 2x SSC for 5 min, rinsed in deionized water and ethanol, and allowed to dry. The sample was covered with 50 µl ice-cold gel solution (4% acrylamide (1610154, Bio-Rad), 2x SSC, 0.2% ammonium persulfate (A3078, Sigma) and 0.2% *N*,*N*,*N'*,*N'*-tetramethylethylenediamine (T7024, Sigma)) and sandwiched by a coverslip functionalized by GelSlick (Lonza, 50640)[17]. The sample was incubated at 4 °C in a homemade nitrogen chamber for 1 h, then for 1.5 h at 37 °C. The coverslip was removed by lifting gently with tweezers from the edge, the sample was incubated in digestion buffer (0.8 M guanidine-HCl (Sigma, G3272), 50 mM Tris-HCl pH 8, 1 mM EDTA, 0.5% (v/v) Triton X-100 in nuclease-free water, 1% (v/v) proteinase K (New England Biolabs, P8107S)) at 100 r.p.m. at 37 °C for 2 h, then washed in 2x SSC twice for 5 min. Encoding and readout then proceeded as in the fifth protocol. Incubations were performed by covering samples on the slide with 100 µl hybridization mix, covering the hybridization mix with a small parafilm square (MilliporeSigma, HS234526B) and storing the slide in a humidity chamber with the same salt concentration as the solution. Washes were performed individually in Petri dishes. Before imaging, gel samples were covered for 5 min in Slowfade mountant (ThermoFisher, S36963) and covered with a small parafilm square.

### Phage infection hybridization

Phage infection cells were stained using the sixth protocol from the preceding section (Supplementary Table 3).

### Spectral and airyscan imaging

Spectral and airyscan images were recorded on an inverted Zeiss 880 confocal microscope equipped with a 32-anode spectral detector, a Plan-Apochromat ×63/1.40 oil objective and excitation lasers at 405 nm, 488 nm, 514 nm, 561 nm and 633 nm using acquisition settings listed in Supplementary Table 4. The microscope was controlled using ZEN v.2.3.

### Manual spot background filtering

Images were processed using a combination of Python scripts using numpy (v.1.21.2)[53] and interactive Jupyter notebooks v.1.0.0 to iteratively adjust and check the results of parameter adjustments. We first applied deconvolution and pixel reassignment to airyscan images to return a super-resolution image using Zen 2.3 SP1 FP3 (Black) v.14.0.28.201. Taking this as input, we then set a manual threshold to identify the foreground. We set the threshold such that visually distinct spots were mostly masked as separate objects. For images with high levels of non-specific signal, 'blobs', we used watershed segmentation with the background thresholded image as seed and a low-intensity background thresholded image as a mask. We measured the foreground objects using scikit-image v.0.17.2 functions. We then removed objects larger than the threshold area. Here we set the threshold such that objects containing 1–3 neighbouring spots were not removed, but objects with the continuous high signal indicative of non-specific binding were removed. We then filtered the remaining objects on the basis of maximum intensity. Here we set the threshold to remove objects with continuous low intensity but kept objects with high-intensity peaks.

### Semi-automated image segmentation

For batches of images, an example image was selected and a zoom region within the image was selected to manually adjust segmentation

parameters. In airyscan images, segmentation parameters were set separately for cell and spot channels. In spectral images, the channels were aligned using phase cross correlation to correct for drift while switching between lasers, then the maximum projection or sum projection along the channel axis was used for segmentation. The image background mask was determined by applying a manual threshold, loading a manually adjusted background mask (as in some spot segmentation), or $k$-means clustering of pixel intensities. For segmentation preprocessing, images were optionally log normalized to enhance dim cells, then denoised using Chambolle total variation denoising implemented in skimage with adjustments to the weight parameter[58,59]. In airyscan images, it was sometimes necessary to blur subcellular features, so a Gaussian filter could be applied with adjustments to the sigma parameter. If objects were densely packed and edge enhancement was required, we applied the local neighbourhood enhancement algorithm to generate an edge-enhanced mask[8]. In certain cases, difference of Gaussians was also used for edge enhancement of the preprocessed image. We then used the watershed algorithm with peak local maxima as seeds to generate the final segmentation. Once the parameters were set, a Snakemake pipeline applied the segmentation parameters to all images in the batch. Segmented objects were measured using standard skimage functions. For spot images, local maxima were determined using skimage functions and objects with multiple local maxima were split into new objects using Pysal[60] to generate a Voronoi diagram from the maxima to set borders between the new objects. Spots were assigned to cells on the basis of object overlap or by radial distance between centroids.

### Spot subcellular location calculation and projection onto density map
For each spot paired with a cell, we calculated $x,y$ coordinates where the $x$ axis is the direction of the cell's long axis, the $y$ axis is the direction of the short axis and the magnitude of each coordinate was normalized to the average cell length and width.

$$x_{\text{spot}} = d_{\text{centroid–spot}} \times \cos\left(\theta_{\text{cell–spot}}\right) \times \frac{\text{length}_{\text{average}}}{\text{length}_{\text{cell}}} \quad (1)$$

$$y_{\text{spot}} = d_{\text{centroid–spot}} \times \sin(\theta_{\text{cell–spot}}) \times \frac{\text{width}_{\text{average}}}{\text{width}_{\text{cell}}} \quad (2)$$

where $d_{\text{centroid–spot}}$ is the distance between the centroid of the cell and the spot, and $\theta_{\text{cell–spot}}$ is the angle between the cell's long axis and the spot-centroid axis. We then created a grid of points to cover the average cell length and width, used the scikit-learn nearest neighbours algorithm to calculate the number of spots within a certain radius of each grid point and divided this number by the area of the search to get a density value for each point.

### Manual cell and spot counting
In the 30 min and 40 min timepoints of the phage infection, many of the infected cells had reduced 16S rRNA signal and lysed cells had caused clumps of cells to form, resulting in difficulties in segmentation. To count cells and classify them by their number of phage spots, we used a manual counting strategy where each image was loaded into a graphic design tool (Affinity Designer) and cells of each type were counted and marked by hand. We counted a minimum of 1,000 cells for each time–MOI combination.

### Prediction of phage infection rates
We used the probability mass function for a Poisson random variable to predict the fraction of cells that would encounter at least one phage

$$f(x) = \frac{e^{-\lambda}\lambda^x}{x!} \quad (3)$$

$$f(x > 0) = 1 - f(0) = 1 - e^{-\lambda} \quad (4)$$

where $x$ is the number of phage a cell collides with and $\lambda$ is the ratio of average phage concentration to average cell concentration (multiplicity of infection).

### Manual seeding of transformed *E. coli* onto plaque samples
Fragments of plaque were aspirated in 50% ethanol storage solution using a 20 µl pipette with a cut tip with a wide bore, deposited on a microscope slide and allowed to dry. We then deposited 2 µl of transformed *E. coli* with induced *GFP* directly on top of the plaque and allowed the slide to dry. We then proceeded through the finalized MGE-FISH method.

### DNA extraction
DNA was extracted from Cornell volunteer plaque samples using the UCP pathogen mini kit (Quiagen, 50214, 19091). DNA was extracted from Harvard School of Dental Medicine patient plaque samples using a modified version of an enzyme-based process[61]. Plaque samples were suspended in 1 ml of TE10 (10 mM Tris-HCl, 10 mM EDTA) buffer. Lysozyme (ThermoFisher, 89833) was added at a final concentration of 15 mg ml$^{-1}$ and the suspension was incubated for 1 h at 37 °C with gentle mixing. Purified achromopeptidase (Wako Pure Chemical) was added at a final concentration of 2,000 units ml$^{-1}$ and the suspension was further incubated for 30 min at 37 °C. Sodium dodecyl sulfate (final concentration, 1 mg ml$^{-1}$) and proteinase K (final concentration, 1 mg ml$^{-1}$; NEB, P8107S) were added to the sample and the mixture was incubated for 1 h at 55 °C. DNA was extracted with phenol/chloroform/isoamyl alcohol (25:24:1), precipitated with isopropanol and 3 M sodium acetate, washed with 75% ethanol and resuspended in 200 µl of TE buffer.

### Sequencing
For short-read sequencing, the purified DNA was fragmented and prepared as an Illumina library (Illumina, FC-131-1096) and sequenced on an Illumina NextSeq 2K with P2 2×100 paired-end reads. For long-read sequencing, the purified DNA was prepared with a Rapid PCR Barcoding kit (Nanopore, SQK-RPB114.24) and sequencing was performed on a Nanopore MinION Mk1B with an R10.4.1 flowcell.

### AMR and prophage gene discovery
Raw reads were processed with PRINSEQ lite (v.0.20.4)[62] and trimmomatic (v.0.36)[63] to remove optical duplicates and sequencing adapters. Reads mapping to the human genome were discarded using BMTagger (Rotmistrovsky, K. and Agarwala, R., unpublished). Clean reads were assembled using SPAdes v.3.14.0 (paired-end mode and –meta option)[64] and reads were aligned to contigs using minimap2 (v.2.17)[65]. Contigs were resolved into metagenomic bins using vamb (v.3.0.2)[33] with reduced hyperparameters (-l 24, -n 384 384). Completeness and contamination of bins were evaluated with checkM (v.1.1.2)[66] and taxonomies were assigned to bins using GTDB-Tk v.1.0.2 with GTDB (release 207)[34,67]. Read-level taxonomic relative abundance estimates were carried out with Kraken2 (v.2.1.2)[68] and Bracken (v.2.6.1)[69]. Lytic and lysogenic phage were identified and evaluated for induction using VIBRANT (v.1.2.1)[31] and PropagAtE (v.1.0.0)[32], requiring a minimum length of 5,000 bp and at least 10 open reading frames per scaffold. Antibiotic resistance genes were annotated on contigs and mobile elements using Resistance Gene Identifier v.5.2.0 against the CARD database v.3.1.0 supplemented with the Resistomes and Variants dataset (v.3.0.8)[70].

### Plasmid prediction
Long raw data was processed using Dorado v.0.4.2. Long reads were assembled using Flye (v.2.9.2)[71]. Hybrid metagenomic assembly was performed using OPERA-MS on clean short reads and Dorado duplex outputs for long reads[72]. Plasmids were predicted using geNomad

v.1.7.1 and annotated with Bakta (v.1.8.1)[73,74]. Putative plasmids from the hybrid assembly were identified in the long read-only assembly to help with circularization of the sequence. Short reads were aligned to putative plasmid assemblies using bowtie2 (v.2.5.1)[75]. Long reads were aligned using bwa mem v.0.7.17 with Nanopore parameters (-x ont2d) and filtered to remove short partial alignments (identity >80%, query coverage >80%)[76]. Coverage measurements were done with samtools (v.1.18)[77]. GC skew was calculated as $(G_{50bp} - C_{50bp})/(G_{50bp} + C_{50bp})$, where $G_{50bp}$ and $C_{50bp}$ are the number of G and C bases in a 50 bp window, and the location of OriC was estimated visually on the basis of GC skew plotting. The number of GGGG and CCCC stretches in each 50 bp sequence was counted as a 4 bp window at each base; for example, GGGGG contributes two counts.

## Plaque MGE-FISH staining

Plaque samples were stained using the fifth or sixth protocol of 'Single-molecule FISH-transformed *E. coli* hybridization method development protocols' with some modifications. Plaque was deposited on a microscope coverslip by aspirating 2 µl of settled plaque gently from the bottom of a plaque sample collection tube with a wide bore pipette tip, depositing on the slide and allowing excess liquid to dry. Cells were then fixed by covering with 2% formaldehyde for 10 min at room temperature, washed for 5 min in 1 M Tris-HCl pH 7.5 for 5 min and washed in 10 mM Tris-HCl pH 8.0 for 2 min. Melpha X solution (prepared as previously reported)[18] was substituted for Label-X solution. Encoding was altered to 12 h at 46 °C in a different hybridization buffer (15% formamide, 5x SSC, 9 mM citric acid (pH 6.0), 0.1% Tween 20, 50 µg ml$^{-1}$ heparin, 1x Denhardt's solution, 10% dextran sulfate), 20 nM encoding probes (Supplementary Tables 5–8, 11 and 12) and 200 nM EUB338 probes (Supplementary Table 1)[15]. After encoding, samples were washed for 5 min at 46 °C in wash buffer (15% formamide, 5x SSC, 9 mM citric acid (pH 6.0), 0.1% Tween 20, 50 µg ml$^{-1}$ heparin), 15 min at 37 °C in fresh wash buffer and 25 min at room temperature in fresh wash buffer. Readout was performed with a new readout buffer (5x SSC, 0.1% Tween 20, 10% dextran sulfate, 60 nM HCR hairpins, 200 nM EUB338 readout probes). After readout, samples were washed for 5 min at room temperature in 5x SSCT (5x SSC, 0.1% Tween 20), 30 min at room temperature in fresh 5x SSCT twice more, then 5 min in fresh 5x SSCT. Samples were covered with Slowfade mountant before imaging.

## Spatial autocorrelation analysis

A neighbour spatial connectivity matrix was constructed from cell segmentation centroids using a Voronoi diagram algorithm from Pysal v.23.7. Each cell was given a binary mark indicating presence of MGE spot. The weight matrix and marked cells were used in a global Moran's *I* test from Pysal to calculate spot autocorrelation. The measured Moran's *I* value was compared against a simulation-based null model assuming that spots are randomly distributed within the cell space. *P* values were calculated using a two-tailed Monte Carlo test.

## Large-scale spot density plots

After spot segmentation, the universal 16S rRNA signal was used to create a global mask to identify the foreground. For each pixel in the foreground, we used the nearest neighbours algorithm to calculate the number of spots within a certain radius of each grid point and divided this number by the area of the search to get a density value for each point.

## Spatial association measurements

We performed two versions of spot co-localization. First, in a given colour channel, for each spot we used the nearest neighbours algorithm to determine whether there were spots of the other colour(s) within a 0.5 µm radius and calculated the fraction of spots co-localized with each of the other colours on the basis of the number of spots in the

reference channel. We repeated the measurement for each colour channel. In the second version, we overlaid the spots from each channel (labelled as different spot types), divided the image into a grid of squares with 5 µm edges, classified each square on the basis of the number of spot types present, counted the number of squares of each type and normalized this number by the total number of squares with at least one spot type.

## AMR gene distribution measurements

Segmented spots were converted into a point pattern object in the PySAL Python package[60]. Simulations were generated using the PoissonPointProcess function to generate 100 realizations of the point pattern. Nearest neighbour distances were generated from these objects with the nnd function. Histogram values were calculated using 1 µm bins. The cumulative distribution $G(d)$ was calculated using the *G* function in PySAL

$$G(d) = \sum_{i=1}^{n} \frac{\Phi_i^d}{n} \tag{5}$$

$$\Phi_i^d = \begin{cases} 1 & \text{if } d_{\min}(s_i) < d \\ 0 & \text{otherwise} \end{cases} \tag{6}$$

where *d* is distance, *n* is the number of spots and $d_{\min}(s_i)$ is the nearest neighbour distance of spot *i*. The pair correlation $R(d)$ was calculated using the *K* function in PySAL

$$K(d) = \frac{\sum_{i=1}^{n} \sum_{j=1}^{n} \Psi_{ij}(d)}{n\hat{\lambda}} \tag{7}$$

$$\Psi_{ij}(d) = \begin{cases} 1 & \text{if } d_{ij} < d \\ 0 & \text{otherwise} \end{cases} \tag{8}$$

$$R(d) = \frac{1}{2\pi d} \frac{\Delta K(d)}{\Delta d} \tag{9}$$

where $d_{ij}$ is the distance between spots *i* and *j*, and $\hat{\lambda}$ is $\frac{n}{\text{total area}}$, the intensity estimation. We calculated the 95% probability envelopes for histogram values, cumulative distribution and pair correlation at each distance *d* using a two-tailed approach. At a given *d*, we selected the upper envelope value such that 2.5% of simulations had greater values at *d* and the lower envelope value such that 2.5% of simulations had lesser values at *d*.

## Genus-level probe design

We performed full-length 16S rRNA sequencing and taxonomic classification as previously described[8] on the extracted DNA used for metagenomic sequencing in 'DNA extraction'. We searched for previously designed genus-level FISH probe sequences[30] and blasted the probes against our full-length 16S rRNA data using blastn. We filtered results to remove 'non-significant' alignments as defined above in 'DNA-FISH split-probe design', determined the fraction of significant alignments to non-target genera and removed probes with off-target rate greater than 0.1. We then selected 5-bit binary barcodes for each genus to maximize the distance between barcode fluorescent spectra. The distance between sum-normalized arrays of reference spectra was calculated using a 'Euclidean distance of cumulative spectrum' metric[78]. On the basis of the binary barcodes, we concatenated a readout sequence to the 3' end of each probe sequence such that the readout sequence would hybridize the appropriate fluorescent readout probe for the barcode (Supplementary Table 9). For barcodes with multiple colours in the barcode, we created separate probes concatenated with each readout sequence. We created barcodes that used only the

488 nm, 514 nm and 561 nm lasers, thus reserving the 633 nm laser for MGE-FISH and the 405 nm laser for the universal EUB338 16S rRNA stain. For stains where we targeted only 5 genera, we simply used a different fluorophore for each genus probe.

### Combined MGE-FISH and HiPR-FISH staining
Samples were prepared with the sixth protocol in 'Single-molecule FISH-transformed *E. coli* hybridization method development protocols' and as in 'Plaque MGE-FISH staining', except for the hybridization buffer which included 20 nM of pooled genus probes and an EUB 338 probe appended with R9 HiPR-FISH flanking region (Supplementary Tables 1 and 9). The readout buffer included 200 nM of each of the five fluorescent readout probes for the genus encoding plus the R9 fluorescent readout probe for the EUB 338 encoding.

### Pixel-level spectral classification
To classify pixels in the 5-genus experiment (for example, Fig. 3c), we aligned the laser channels of the spectral images using phase cross correlation, then we performed Gaussian blurring (sigma = 3) on each spectral channel to reduce the noise in each pixel's spectra. We acquired a maximum intensity projection along the channel axis, selected a background threshold and generated a mask. To account for non-specific binding, which generated a low-intensity background signal with the '11111' (all 5 fluorophores) spectral barcode, we multiplied the '11111' reference spectrum by a scalar and subtracted the scaled spectrum from each pixel's measured spectrum (reference spectra for each barcode were collected as previously described)[8]. We visualized the pixel spectra before and after subtraction and adjusted the scalar such that the visually apparent background was removed (scalar = 0.05). The adjusted pixel spectra were stored in a 'pixel spectra matrix' with the following shape: number of pixels, number of spectral channels. The reference spectra for all barcodes were sum normalized and merged in a 'reference spectra matrix' with the following shape: number of spectral channels, number of barcodes. We performed matrix multiplication between the 'pixel spectra matrix' and the 'reference spectra matrix' to get a 'classification matrix' with shape: number of pixels, number of barcodes. Separately, we evaluated the reference spectra and created a boolean array indicating whether we expected a signal from each of the three lasers. We merged these arrays into a 'reference laser presence' matrix with shape: number of lasers, number of barcodes. Then, for each adjusted pixel spectrum, we measured the maximum value for each laser, normalized these values by the highest of the three values and set minimum threshold values (threshold$_{488}$ = 0.3, threshold$_{514}$ = 0.4, threshold$_{561}$ = 0.3) to create a 'pixel laser presence' boolean matrix with shape: number of pixels, number of lasers. We performed matrix multiplication between the 'pixel laser presence' matrix and the 'reference laser presence' matrix to get a matrix with shape: number of pixels, number of barcodes. We performed element-wise multiplication between this matrix and the 'classification matrix' to remove barcodes from the classification matrix if the signal from one of the lasers was too low. For each pixel, we selected the barcode with the highest value in the adjusted 'classification matrix'.

### Cell segmentation-level spectral classification
We aligned the laser channels using phase cross correlation, then applied the 'Semi-automated image segmentation' method to the maximum projection of the spectral channels. In the 5-genus experiment, for each object in the cell segmentation, if all the pixels within the object were assigned to the same taxon, we assigned that taxon to the object. If multiple taxa were represented in the cell pixels, the object was split into multiple new objects such that each new object encompassed pixels of only one taxon. To classify segmented cells in the 18-genus experiment (for example, Fig. 3g), We acquired the mean spectrum of pixels within each segmented object, then calculated the pairwise cosine distances between all mean cell spectra and clustered the spectra into 20 groups using agglomerative clustering. We then manually classified each cluster by visually comparing them to pure reference spectra, which we acquired as reported previously[8].

### Registration of airyscan and lambda mode images
Since HiPR-FISH images were captured using lambda mode for spectral measurement and MGE-FISH images were captured using airyscan mode for improved resolution, we rescaled the HiPR-FISH images so that the pixel size matched the MGE-FISH images. We used phase cross correlation to register shifts between the airyscan 16S rRNA signal and the HiPR-FISH maximum projection image. We then applied these shifts to the airyscan MGE-FISH images.

### Taxon–spot spatial association measurements
Given the set of cell centroids and spot coordinates, we used the nearest neighbour algorithm from scikit-learn to identify the nearest cell to each spot[79]. We then calculated the fraction of spots associated with each taxon and the fraction of each taxon associated with spots.

### Random simulation of spot distribution
We used the foreground mask to create a list of pixel coordinates within the plaque cells, then used a random integer generator to select pixels by their list index. We used the randomly selected pixel coordinates as simulated spots and counted taxon–spot spatial associations as described above. This was repeated for 1,000 simulations, and we calculated the mean and standard deviation for the count values for each taxon. We then calculated the $z$-score for the count values: $z$ = (count − mean)/standard deviation. $P$ values were calculated by counting the fraction of simulations with greater values than the observed value.

### Statistics
No statistical methods were used to pre-determine sample sizes, but our sample sizes are similar to those reported in previous publications[8,30]. For cultured cell experiments, the number of cells measured was routinely in the thousands. Sample size was chosen on the basis of fields of view, where each condition was measured with three tile scans composed of four fields of view each, thus measuring thousands of cells. For oral plaque experiments, the target genes were unique to each volunteer, so multiple samples were not possible for a given target gene. Sample size was chosen on the basis of fields of view, where each sample was measured with at least three tile scans composed of at least 4 fields of view each, thus measuring thousands of cells. For technical controls, samples of cultured cells and plaque were allocated randomly. Data collection and analysis were not performed blind to the conditions of the experiments. No data were excluded from the analysis. Python v.3.8.5 was used to generate statistics. Boxplots consist of a bottom line representing the lower quartile (Q1), a line inside the box representing the median (Q2), a top line representing the upper quartile (Q3), an upper whisker extending from the top of the box indicating the maximum value within 1.5 times the interquartile range (IQR) above Q3 and a lower whisker extending from the bottom of the box indicating the minimum value within 1.5 times the IQR below Q1. Monte Carlo methods with 100 or 1,000 simulations were used in two-sided tests to evaluate null hypotheses of random distribution of spots. Data distribution was assumed to be normal, but this was not formally tested.

### Reporting summary
Further information on research design is available in the Nature Portfolio Reporting Summary linked to this article.

## Data availability
Illumina and PacBIO sequencing data are available at the National Center for Biotechnology Information Sequence Read Archive with accession number PRJNA981198. Microscopy data have been deposited

at Zenodo at https://doi.org/10.5281/zenodo.8015720 (Fig. 1b and Extended Data Fig. 1a,b)[80], https://doi.org/10.5281/zenodo.8015754 (Fig. 1c and Extended Data Fig. 1c,d, including count tables)[81], https://doi.org/10.5281/zenodo.8015832 (Fig. 2 and Extended Data Figs. 2 and 3)[82], https://zenodo.org/doi/10.5281/zenodo.11039333 (Fig. 3 and Extended Data Fig. 4, including plasmid assembly with Illumina and Nanopore reads)[83] and https://zenodo.org/doi/10.5281/zenodo.11039443 (Fig. 4 and Extended Data Fig. 5, including plasmid assembly with Illumina and Nanopore reads)[84].

GTDB release 207 is available at https://data.gtdb.ecogenomic.org/releases/release207/, CARD v.3.1.0 is available at https://card.mcmaster.ca/download, PLSDB v.2023_11_03 is available at https://ccb-microbe.cs.uni-saarland.de/plsdb/plasmids/download/, RefSeq release 220 is available at https://ftp.ncbi.nlm.nih.gov/refseq/release/release-catalog/archive/, checkM database v.2015-01-16 is available at https://zenodo.org/doi/10.5281/zenodo.7401544 (ref. 85), geNomad database v.1.7 is available at https://doi.org/10.5281/zenodo.10594875 (ref. 86) and the Bakta database v.5.0 is available at https://doi.org/10.5281/zenodo.7669534 (ref. 87). For VIBRANT, Pfam v.32.0 is available https://ftp.ebi.ac.uk/pub/databases/Pfam/releases/Pfam32.0/, VOG v.94 is available at https://fileshare.lisc.univie.ac.at/vog/vog94/ and KEGG v.2019-03-20 is available at ftp://ftp.genome.jp/pub/db/kofam/archives/2019-03-20/ (ref. 31).

## Code availability

The specific implementation of code to generate figures presented here is available on GitHub at https://github.com/benjamingrodner/hipr_mge_fish (v.1.0.0, https://zenodo.org/10.5281/zenodo.11085744)[88]. The generalized pipeline for segmentation is available at https://github.com/benjamingrodner/pipeline_segmentation (v.1.0.0, https://doi.org/10.5281/zenodo.11085837)[89], while the generalized implementation of probe design is available at https://github.com/benjamingrodner/FISH_split_probe_design (v.1.0.0, https://doi.org/10.5281/zenodo.11085839)[90].

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

## Acknowledgements

We thank R. M. Williams and J. M. Dela Cruz for assistance with microscopy, T. Doerr (Weill Institute for Cell and Molecular Biology, Cornell University, Ithaca, New York) for providing materials, and M. Mantri, D. W. McKellar, J. Jones, L. Takayasu, S. Arias and T. Ciavatti for discussions and feedback. This work was supported by an instrumentation grant from the Kavli Institute at Cornell, by US National Institutes of Health (NIH) grants 1DP2AI138242 to I.D.V. and 1R33CA235302 to I.D.V., W.R.Z. and I.L.B, and by the 2022 Osseointegration Foundation Basic Science Research Grant. Imaging data were acquired in the Cornell Biotechnology Resource Center Imaging Facility using the shared, NYSTEM (CO29155)- and NIH (S10OD018516)-funded Zeiss LSM880 confocal and multiphoton microscope.

## Author contributions

B.G., H.S. and I.D.V. conceived the study. B.G., H.S., O.F., I.N., A.C.V., P.J.D., W.R.Z., I.L.B. and I.D.V. designed staining and imaging methods and validation experiments. B.G., O.F. and I.N. performed sequencing, staining, and imaging methods and validation experiments. B.G., H.S., D.T.W., C.-Y.C. and D.M.K. collected human samples. D.T.W., C.-Y.C. and D.M.K. conceived the patient experiments. A.C.V. analysed metagenomic sequencing data to identify target genes and designed and performed the in vitro phage infection system. P.J.D. conceived, designed and performed the in vitro plasmid system. B.G., H.S. and O.F. wrote the probe design and image analysis pipelines. B.G. and I.D.V. wrote the paper and prepared the figures. All authors read and edited the paper.

## Competing interests

H.S. is a co-founder at Kanvas Biosciences. I.D.V. is a member of the Scientific Advisory Board of Karius Inc. and GenDX, and a co-founder of Kanvas Biosciences. H.S. and I.D.V. are listed as inventors on patents related to multiplexed imaging methods (US20210047634A1, United States, 2019; US20230159989A1, United States, 2022; US20230265504A1, United States, 2023). P.J.D. is an employee of Kanvas Biosciences. The other authors declare no competing interests.

## Additional information

**Extended data** is available for this paper at https://doi.org/10.1038/s41564-024-01735-5.

**Correspondence and requests for materials** should be addressed to Iwijn De Vlaminck.

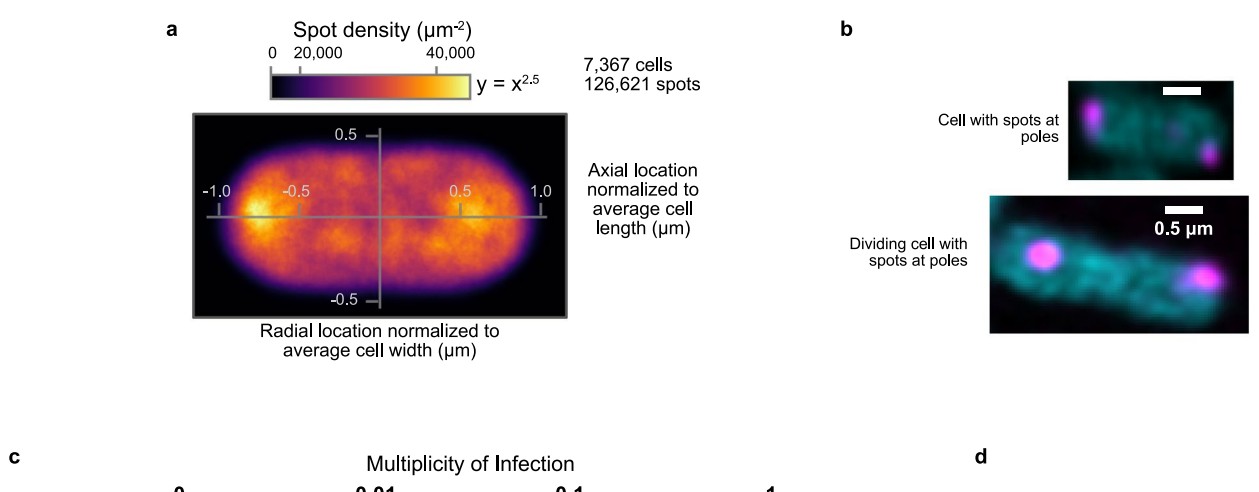

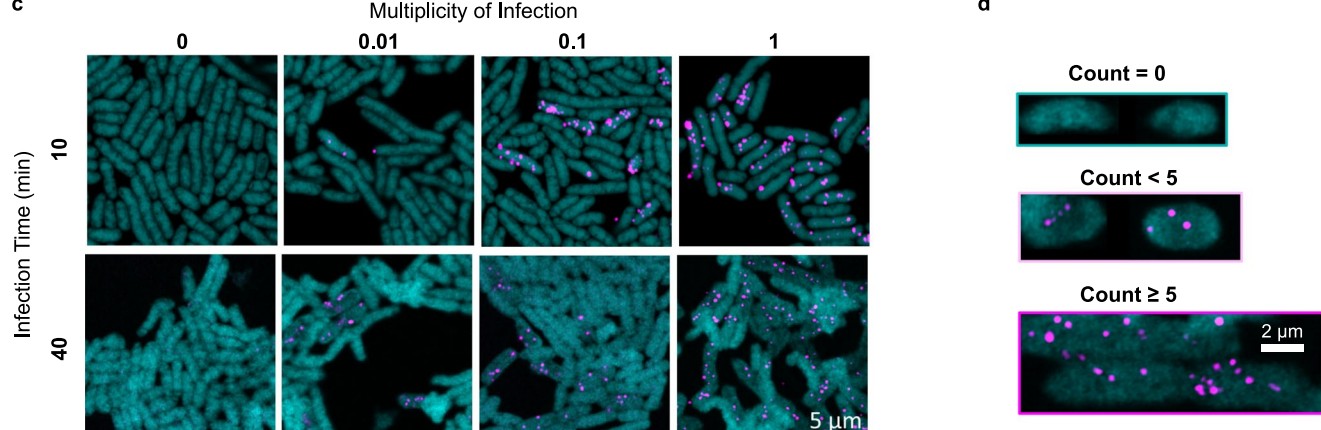

**Extended Data Fig. 1 | In vitro single MGE-FISH on plasmids and phage.**
**a** Subcellular spot locations MGE-FISH staining of *GFP*-plasmid normalized to average cell shape plotted as spot density. The colormap is nonlinear and maps to values using a power law where color value $x$ maps to density value $y = x^{2.5}$. **b** Example zoomed raw images of cells where spots are located at the poles.

**c** Example images of MGE-FISH staining of T4 phage *gp34* gene in T4 phage infection time series. Cyan: 16S rRNA, magenta: GFP DNA. **d** Example cells showing examples for the different classification of cells in the manual counting data in Fig. 1c.

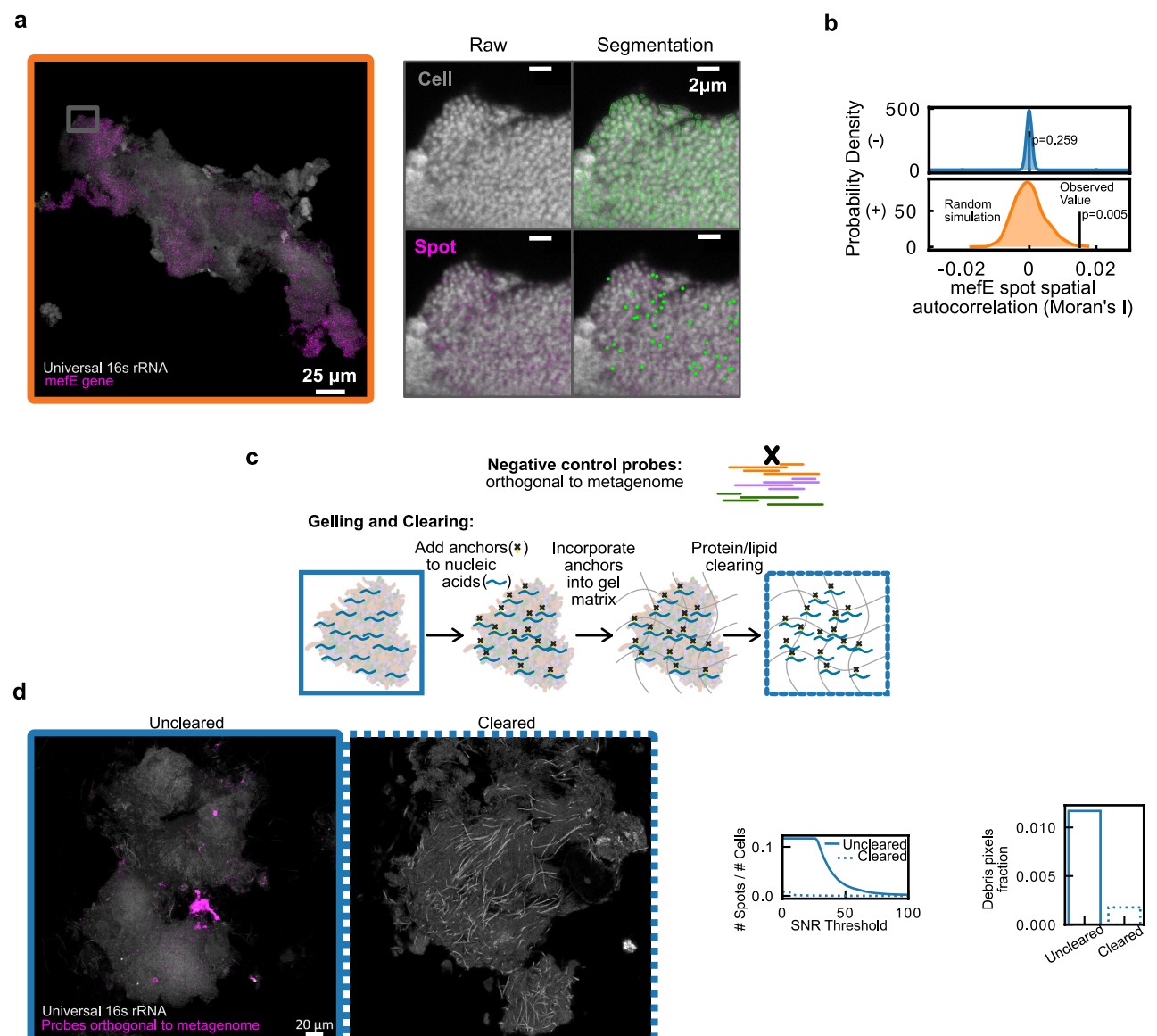

**Extended Data Fig. 2 | Evaluation of MGE-FISH in plaque biofilms.**
**a** Visualization of cell segmentation and spot counting on a zoom region (gray box on the *left*) of Fig. 2c. Green outlines in *top right* image indicate cell segmentations and green dots on the *bottom right* image indicate spot locations. **b** Spatial autocorrelation of *mefE* spots using Moran's I statistic. Colored vertical bar indicates the observed value, black vertical bar indicates the mean value of the simulation, and the shaded area indicates the histogram of the simulation. For the simulation spots were randomly redistributed on the same set of cell segmentations 1000 times. The Monte Carlo method with 1000 simulations was used in a two-sided test to evaluate the null hypotheses of random distribution of spots. **c** *Top*: diagram of orthogonal control probes that should produce no signal. *Bottom*: diagram of the gel embedding, nucleic acid anchoring, and sample clearing process. **d** *Left*: Example images showing the off-target signal from orthogonal probes in uncleared and cleared plaque samples. At least three tiled fields of view were collected for each sample with similar results. *Center*: spot counts normalized by number of cells as a function of signal to noise ratio (SNR). *Right*: Measurement of non-spot pixels normalized by cell pixels.

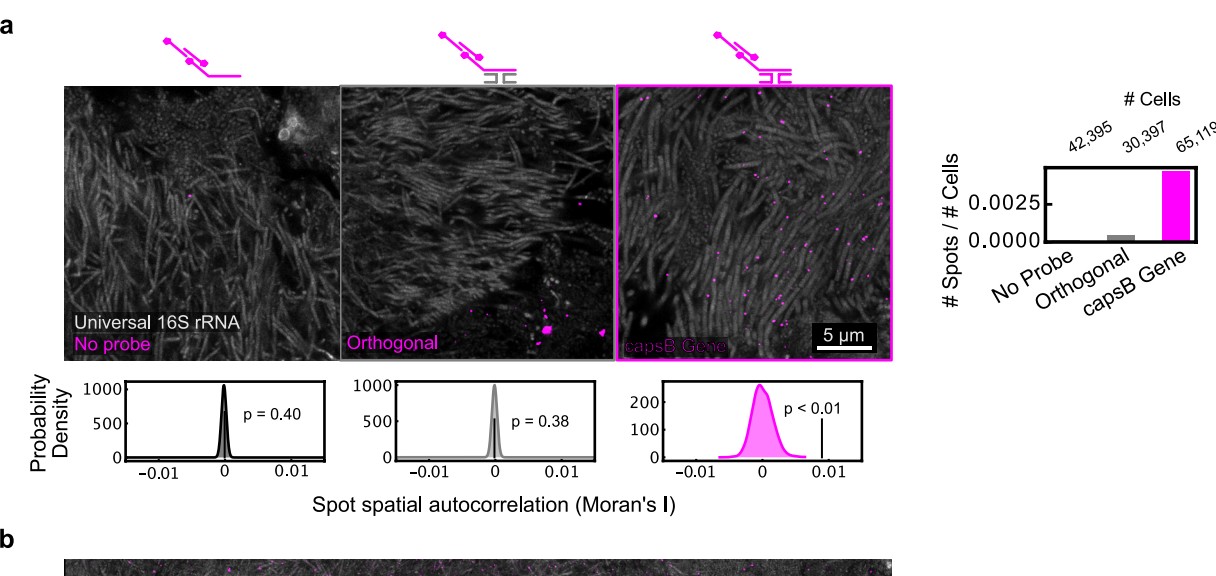

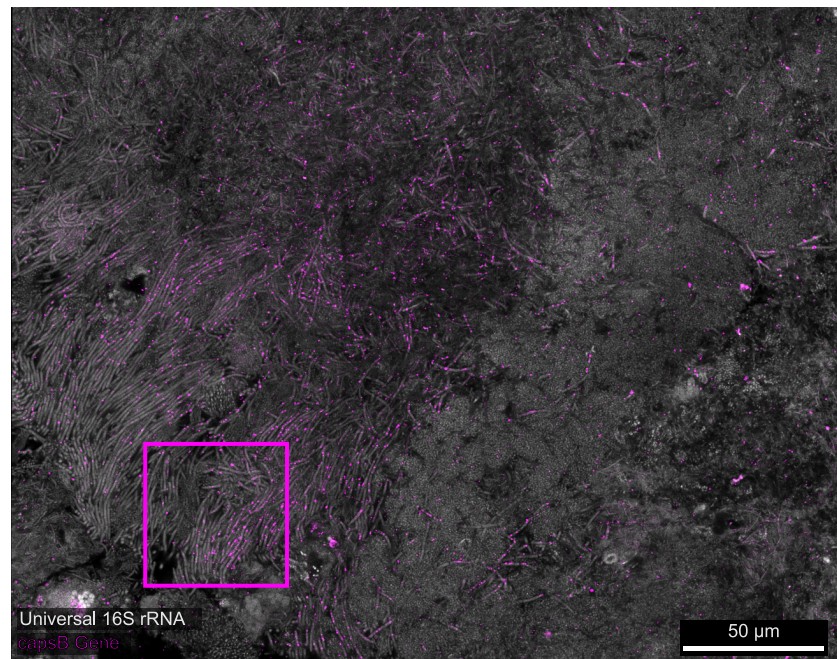

**Extended Data Fig. 3 | Control experiments for T7-like prophage capsB minor capsid protein in Plaque. a** *Top left*: example images showing regions of cells with similar morphology cells. From right to left the images are: MGE-FISH controls with no encoding probes, encoding probes that are orthogonal as determined by metagenomic analysis, or probes targeting *capsB*. *Top right*: spot counts from each control normalized by number of cells. *Bottom*: observed Moran's I spatial autocorrelation values (vertical black lines) compared to 999 simulations of random spot distribution (filled curves). The Monte Carlo method with 999 simulations was used in a two-sided test to evaluate the null hypotheses of random distribution of spots. P-values were 0.40, 0.38, and <0.01 for images captured using no encoding probes, orthogonal encoding probes, and *capsB* encoding probes respectively. **b** Example FOV showing a large hotspot of prophage (~100µm). Inset square shows the location of the *capsB* example image in **a**.

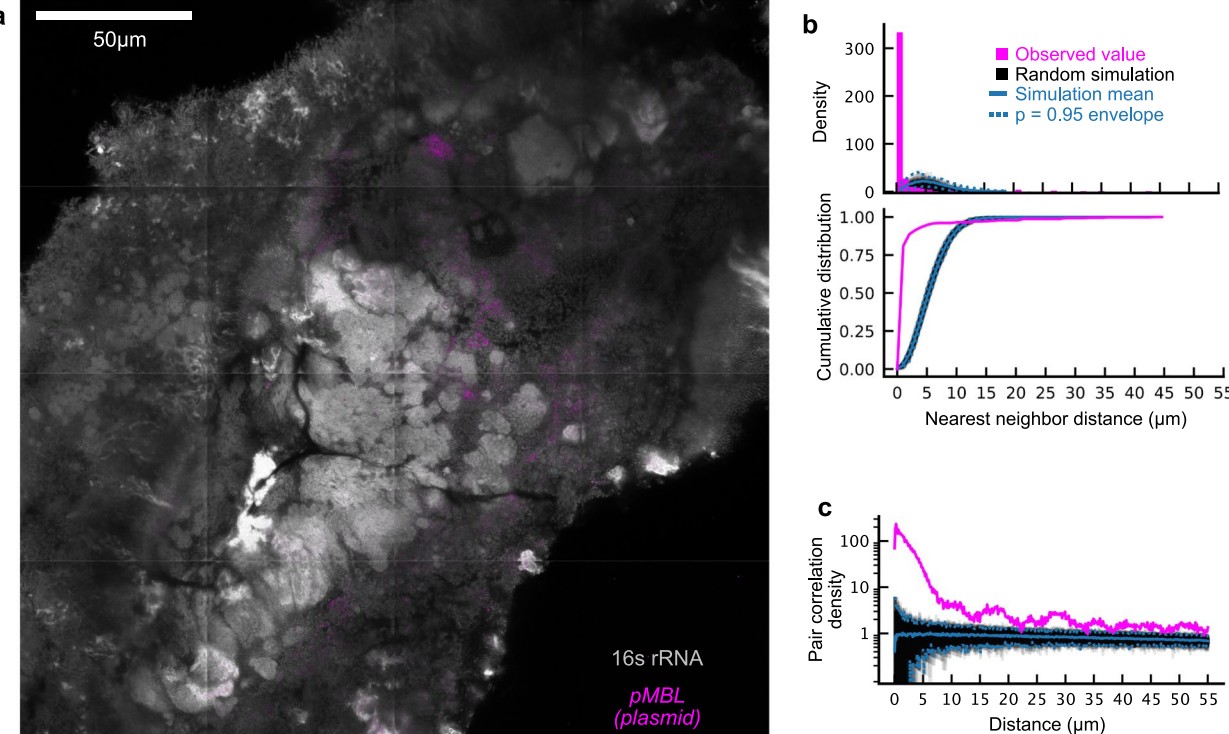

**Extended Data Fig. 4 | AMR gene distribution measurements. a** Example image showing spatially clustered signal from an AMR gene, *pMBL*, found on a plasmid in the metagenomic sequencing data. *pMBL* signal is in magenta and the 16 s rRNA signal is in gray. **b** *Top:* Histograms of *pMBL* nearest neighbor distances for the observed *pMBL* spots (magenta) in **a** and 100 simulations of randomly distributed *pMBL* spots (black). The solid blue line shows the mean of the simulated histograms. 97.5% of simulation values were less than the top dotted blue line, and 97.5% of simulation values were greater than the bottom dotted blue line. The Monte Carlo method with 100 simulations was used in a two-sided test to evaluate the null hypotheses of random distribution of spots. *Bottom:* Empirical probability that a *pMBL* spot has a nearest neighbor distance less than the given distance for observed *pMBL* spots (magenta) and simulated random *pMBL* spots (black). Blue solid and dashed lines are plotted as above. **c** Empirical pair correlation function for observed (magenta) and simulated (black) *pMBL* spots. Values indicate the radial density of *pMBL* spots at a given distance from a reference spot. Blue solid and dashed lines are plotted as above.

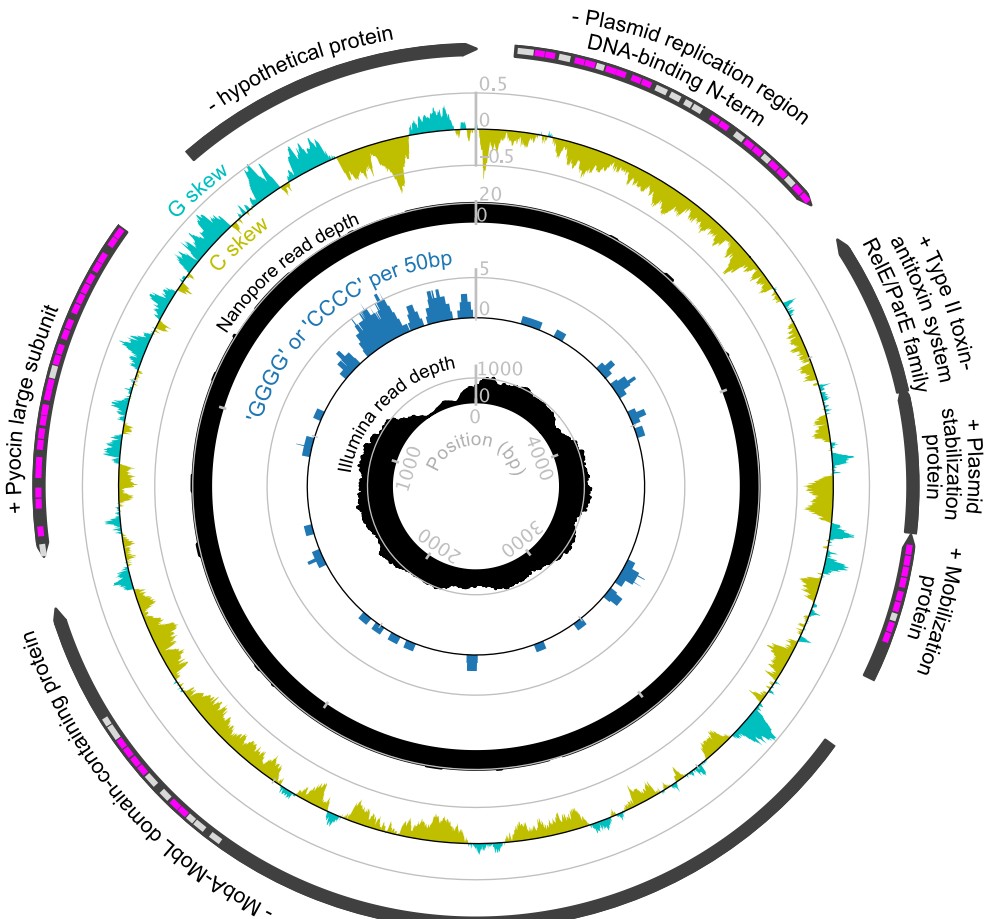

**Extended Data Fig. 5 | Diagram of the previously undescribed plasmid.** Same as Fig. 4b but with additional information. The second ring from the inside indicates in blue the number of tetramers of G or C nucleotide bases in the assembled sequence per 50 base pair window. The second ring from the outside indicates the GC skew in a 50 bp window where 0 indicates equal counts of G and C bases, positive indicates excess G bases, and negative indicates excess C bases.

# Reporting Summary

## Statistics

For all statistical analyses, confirm that the following items are present in the figure legend, table legend, main text, or Methods section.

| n/a | Confirmed | |
|---|---|---|
| ☐ | ☒ | The exact sample size (*n*) for each experimental group/condition, given as a discrete number and unit of measurement |
| ☐ | ☒ | A statement on whether measurements were taken from distinct samples or whether the same sample was measured repeatedly |
| ☐ | ☒ | The statistical test(s) used AND whether they are one- or two-sided<br>*Only common tests should be described solely by name; describe more complex techniques in the Methods section.* |
| ☒ | ☐ | A description of all covariates tested |
| ☐ | ☒ | A description of any assumptions or corrections, such as tests of normality and adjustment for multiple comparisons |
| ☐ | ☒ | A full description of the statistical parameters including central tendency (e.g. means) or other basic estimates (e.g. regression coefficient) AND variation (e.g. standard deviation) or associated estimates of uncertainty (e.g. confidence intervals) |
| ☐ | ☒ | For null hypothesis testing, the test statistic (e.g. *F*, *t*, *r*) with confidence intervals, effect sizes, degrees of freedom and *P* value noted<br>*Give P values as exact values whenever suitable.* |
| ☒ | ☐ | For Bayesian analysis, information on the choice of priors and Markov chain Monte Carlo settings |
| ☒ | ☐ | For hierarchical and complex designs, identification of the appropriate level for tests and full reporting of outcomes |
| ☒ | ☐ | Estimates of effect sizes (e.g. Cohen's *d*, Pearson's *r*), indicating how they were calculated |

*Our web collection on statistics for biologists contains articles on many of the points above.*

## Software and code

Policy information about availability of computer code

| | |
|---|---|
| Data collection | Confocal microscope images were collected on Zeiss LSM i880 with Zen 2.3 SP1 FP3 (Black) v14.0.28.201 software. Nanopore reads were collected on MinION Mk1B with MinKNOW v23.07.15. |
| Data analysis | DNA-FISH Split-Probe design. Probes were designed using a custom Snakemake v7.18.2 pipeline with rules written in Python v3.6.8 using numpy v1.15.4 and pandas v0.24.1.53, Target gene sequences were taken as inputs along with a reference blast database. The target was aligned to the blast database and all significant alignments were recorded for future filtering (blastn v2.13.0). All possible oligonucleotide probes were designed to be complementary to the coding strand of the target gene (i.e. the same sense as the mRNA) using Primer3 v2.3.5.55 Pairs of Probes in this pool were identified as any probes aligning less than three base pairs distant from each other. These probe pairs were then blasted against the reference database using blastn from NCBI. On-target blast results were removed from the results using the target gene alignment IDs. Non-significant blast results were then filtered using user-defined parameters. These include maximum continuous homology (12), GC count (7), and melting temperature (46℃). All blast results with values in these parameters that were less than the specified thresholds were removed as "non-significant alignments". The remaining blast results were considered "significant" or likely to produce off-target signal. Probe pairs were removed when both probes had off-target homologies to nearby regions in the reference database. This nearness parameter is another user-defined threshold. The remaining probe pairs were then sorted with favored probes having low levels of off-target homology. Going down the sorted list, probe pairs were then selected to tile along the gene without overlapping. Selected probes were then appended with appropriate flanking regions so that the target would be stained with the intended fluorophore (Supp. Tab. 1). Two base-pair spacers nucleotides between the flanking region and the probe were selected to minimize the off-target homology of the full-length probes in a similar manner to how probe pairs were sorted by blast results. The pool of selected probe pairs was then evaluated by searching for any off-target homologies where two probes were nearby each other. "Helper" probes were then selected from the Primer3 to tile along the gene without overlapping the existing probes. The final probes were then submitted for oligo synthesis to Integrated DNA Technologies (IDT) at a concentration of 200μM. |

DNA-FISH single probe design. Single probes were designed much as the split probes up to the Primer3 step. Then, instead of pairing probes, the probes were all blasted against the database and the blast results were filtered as the split probes were for "significant" off-target homologies. Probes with any significant off-target homologies were removed and the remaining probes were tiled along the target gene to ensure no overlap. The selected probes were then paired with flanking regions for the readout stain and two base pair spacers were added and optimized as in the split probe design. The resulting probes were submitted for synthesis to IDT.

Orthogonal probe design. Probes with zero significant off-target blasts were selected from split probe pairs for different genes. For example if the left probe from a pair targeting Gene A has zero off-target blasts it is selected, then the right probe from a pair targeting Gene B is selected. The concept is that it is very unlikely these probes will hybridize close enough to each other to initiate HCR fluorescence amplification. Three right probes and three left probes

Manual spot background filtering. Images were processed using a combination of Python scripts using numpy v1.21.2 and interactive Jupyter notebooks v1.0.0 to iteratively adjust and check the results of parameter adjustments. We first applied deconvolution and pixel reassignment to Airyscan images to return a super resolution image using Zen 2.3 SP1 FP3 (Black) v14.0.28.201. Taking this as input, we then set a manual threshold to identify the foreground. We set the threshold such that visually distinct spots were mostly masked as separate objects. For images with high levels of non-specific signal, "blobs", we used watershed segmentation with the background thresholded image as seed and a low intensity background thresholded image as a mask. We measured the foreground objects using skimage functions. We then removed objects larger than the threshold area. Here we set the threshold such that objects containing 1-3 neighboring spots were not removed, but objects with the continuous high signal indicative of non-specific binding were removed. We then filtered the remaining objects based on maximum intensity. Here we set the threshold to remove objects with continuous low intensity, but keep objects with high intensity peaks.

Semi-automated image segmentation. For batches of images, an example image was selected and a zoom region within the image was selected to manually adjust segmentation parameters. In Airyscan images, segmentation parameters were set separately for cell and spot channels. In spectral images, the channels were aligned using phase cross correlation to correct for drift while switching between lasers, then the maximum projection or sum projection along the channel axis was used for segmentation. The image background mask was determined by applying a manual threshold, loading a manually adjusted background mask (as in some spot segmentation), or k-means clustering of pixel intensities. For segmentation pre-processing, images were optionally log normalized to enhance dim cells, then denoised using Chambolle total variation denoising implemented in skimage with adjustments to the weight parameter. In airyscan images it was sometimes necessary to blur subcellular features, so a gaussian filter could be applied with adjustments to the sigma parameter. If objects were densely packed and edge enhancement was required, we applied the local neighborhood enhancement algorithm to generate an edge-enhanced mask.8 In certain cases, difference of gaussians was also used for edge enhancement of the preprocessed image. We then used the watershed algorithm with peak local maxima as seeds to generate the final segmentation. Once the parameters were set, a Snakemake pipeline applied the segmentation parameters to all images in the batch. Segmented objects were measured using standard skimage functions. For spot images, local maxima were determined using skimage functions and objects with multiple local maxima were split into new objects using Pysal60 to generate a Voronoi diagram from the maxima to set borders between the new objects. Spots were assigned to cells based on object overlap or by radial distance between centroids.

Manual Cell and spot counting. In the 30 minute and 40 minute timepoints of the phage infection, many of the infected cells had reduced 16S rRNA signal and lysed cells had caused clumps of cells to form that were difficult to segment. To count cells and classify them by their number of phage spots we used a manual counting strategy where each image was loaded into a graphic design tool (Affinity Designer) and cells of each type were counted and marked by hand. We counted a minimum of 1000 cells for each time-MOI combination.

AMR and prophage gene discovery. Raw reads were processed with PRINSEQ lite v0.20.4 and trimmomatic v0.36 to remove optical duplicates and sequencing adapters. Reads mapping to the human genome were discarded using BMTagger. Clean reads were assembled using SPAdes v3.14.0 (paired-end mode and –meta option) and reads were aligned to contigs using minimap2 v2.17. Contigs were resolved into metagenomic bins using vamb v3.0.2 with reduced hyperparameters (-l 24, -n 384 384). Completeness and contamination of bins were evaluated with checkM v1.1.2, and taxonomies were assigned to bins using GTDB-Tk v1.0.2 with GTDB release 207. Read-level taxonomic relative abundance estimates were carried out with Kraken2 v2.1.2 and Bracken v2.6.1. Lytic and lysogenic phage were identified and evaluated for induction using VIBRANT v1.2.1 and PropagAtE v1.0.0, requiring a minimum length of 5000 bp and at least 10 ORFs per scaffold. Antibiotic resistance genes were annotated on contigs and mobile elements using Resistance Gene Identifier v5.2.0 against the CARD database v3.1.0 supplemented with the Resistomes & Variants dataset v3.0.8.

Plasmid prediction. Long raw data was processed using Dorado v0.4.2. Long reads were assembled using Flye v2.9.2. Hybrid metagenomic assembly was performed using OPERA-MS on clean short reads and Dorado duplex outputs for long reads. Plasmids were predicted using geNomad v1.7.1. Putative plasmids from the hybrid assembly were identified in the long read-only assembly to help with circularization of the sequence. Short reads were aligned to putative plasmids assemblies using bowtie2 v2.5.1. Long reads were aligned using bwa mem v0.7.17 with Nanopore parameters (-x ont2d) and filtered to remove short partial alignments (identity > 80%, query coverage > 80%). Coverage measurements were done with samtools v1.18. GC skew was calculated as $(G50bp - C50bp)/(G50bp + C50bp)$ where G50bp and C50bp are the number of G and C bases in a 50bp window, and the location of OriC was estimated visually based on GC skew plotting. The number of GGGG and CCCC stretches in a plasmid sequence was counted as a 4bp window at each base; for example, GGGGG results in two counts.

Spatial autocorrelation analysis. A neighbor spatial connectivity matrix was constructed from cell segmentation centroids using a Voronoi diagram algorithm from Pysal Each cell was given a binary mark indicating presence of MGE spot. The weight matrix and marked cells were used in a global Moran's I test from Pysal to calculate spot autocorrelation. The measured Moran's I value was compared against a simulation based null model that spots are randomly distributed within the cell space. P-values were calculated using a two tailed Monte Carlo test.

Large scale spot density plots. After spot segmentation, the universal 16S rRNA signal was used to create a global mask to identify the foreground. For each pixel in the foreground, we used the nearest neighbors algorithm to calculate the number of spots within a certain radius of each grid point, and divided by the area of the search to get a density value for each point.

Spatial association measurements. We performed two versions of spot colocalization. First in a given color channel, for each spot we used the nearest neighbors algorithm to determine whether there were spots of the other color(s) within a 0.5μm radius and calculated the fraction of spots colocalized with each of the other colors based on the number of spots in the reference channel. We repeated the measurement for each color channel. In the second version, we overlaid the spots from each channel (labeled as different spot types), divided the image into a grid of squares with 5μm edges, classified each square based on the number of spot types present, counted the number of squares of each

type, and normalized by the total number of squares with at least one spot type.

AMR gene distribution measurements. Segmented spots were converted into a point pattern object in the PySAL python package. Simulations were generated using the PoissonPointProcess function to generate 100 realizations of the point pattern. Nearest neighbor distances were generated from these objects with the nnd function. Histograms values were calculated using 1μm bins. The cumulative distribution G(d) was calculated using the G function from PySAL. The pair correlation function R(d) was calculated using the K function from PySAL.

Genus level probe design. We performed full length 16S rRNA sequencing and taxonomic classification as previously described[8] on the extracted DNA used for metagenomic sequencing in DNA Extraction. We searched for previously designed genus level FISH probe sequences[29] and blasted the probes against our full length 16S rRNA data using blastn. We filtered results to remove "non-significant" alignments as defined above in DNA-FISH Split-Probe design, determined the fraction of significant alignments to non-target genera, and removed probes with off-target rate greater than 0.1. We then selected 5-bit binary barcodes for each genus to maximize the distance between barcode fluorescent spectra. Distance between sum-normalized arrays of reference spectra was calculated using a "e uclidean distance of cumulative spectrum" metric.[77] Based on the binary barcodes we concatenated a readout sequence to the three prime end of each probe sequence such that the readout sequence would hybridize the appropriate fluorescent readout probe for the barcode (Supp. Tab. 9). For barcodes with multiple colors in the barcode, we created separate probes concatenated with each readout sequence. We created barcodes that used only the 488 nm, 514 nm, and 561 nm lasers, thus reserving the 633 nm laser for MGE-FISH and the 405 nm laser for the universal EUB338 16S rRNA stain. For stains where we targeted only 5 genera, we simply used a different fluorophore for each genus probe.

Pixel-level spectral classification. To classify pixels in the 5 genus experiment (e.g. Fig. 3c), we aligned the laser channels of the spectral images using phase cross correlation, then we performed gaussian blurring (sigma=3) on each spectral channel to reduce the noise in each pixel's spectra. We acquired a maximum intensity projection along the channel axis, selected a background threshold, and generated a mask. To account for nonspecific binding, which generates a low intensity background signal with the "11111" (all 5 fluorophores) spectral barcode, we multiplied the "11111" reference spectrum by a scalar and subtracted the scaled spectrum from each pixel's measured spectrum (reference spectra for each barcode were collected as previously described). We visualized the pixel spectra before and after subtraction and adjusted the scalar such that the visually apparent background was removed (scalar=0.05). The adjusted pixel spectra were stored in a "pixel spectra matrix" with the following shape: (number of pixels,number of spectral channels). The reference spectra for all barcodes were sum normalized and merged in a "reference spectra matrix" with the following shape: (number of spectral channels,number of barcodes). We performed matrix multiplication between the "pixel spectra matrix" and the "reference spectra matrix" to get a "classification matrix" with shape: (number of pixels,number of barcodes). Separately, we evaluated the reference spectra and created a boolean array indicating whether or not we expected a signal from each of the three lasers. We merged these arrays into a "reference laser presence" matrix with shape: (number of lasers,number of barcodes). Then, for each adjusted pixel spectrum we measured the maximum value for each laser, normalized these values by the highest of the three values, and set minimum threshold values (threshold488=0.3, threshold514=0.4, threshold561=0.3) to create a "pixel laser presence" boolean matrix with shape: (number of pixels,number of lasers). We performed matrix multiplication between the "pixel laser presence" matrix and the "reference laser presence" matrix to get a matrix with shape: (number of pixels,number of barcodes). We performed element-wise multiplication between this matrix and the "classification matrix" to remove barcodes from the classification matrix if the signal from one of the lasers was too low. For each pixel, we selected the barcode with the highest value in the adjusted "classification matrix".

Cell segmentation level spectral classification. We aligned the laser channels using phase cross correlation, then applied the Semi-automated image segmentation method to the maximum projection of the spectral channels. In the 5-genus experiment, for each object in the cell segmentation, if all the pixels within the object were assigned to the same taxon, we assigned that taxon to the object. If multiple taxa were represented in the cell pixels, the object was split into multiple new objects such that each new object encompassed pixels of only one taxon. To classify segmented cells in the 18-genus experiment (e.g. Fig. 3g), We acquired the mean spectrum of pixels within each segmented object, then calculated the pairwise cosine distances between all mean cell spectra and clustered the spectra into 20 groups using agglomerative clustering. We then manually classified each cluster by visually comparing them to pure reference spectra, which we acquired as reported previously.

Registration of Airyscan and Lambda mode images. Since HiPR-FISH images were captured using Lambda mode for spectral measurement and MGE-FISH images were captured using Airyscan mode for improved resolution, we rescaled the HiPR-FISH images so that the pixel size matched the MGE-FISH images. We used phase cross-correlation to register shifts between the Airyscan 16s rRNA signal and the HiPR-FISH maximum projection image. We then applied these shifts to the Airyscan MGE-FISH images.

Taxon-spot spatial association measurements. We created isolated the a subset set of the cell centroids for each taxon. Then for each taxon we used the nearest neighbor algorithm to measure the distance from each spot to the nearest cell of that taxon and counted the number of spots where distance was less than 0.5μm. To calculate the fraction of spots and taxon cells, we divided the count by the total number of spots and total number of taxon cells respectively.

Random simulation of spot distribution. We used the foreground mask to create a list of pixel coordinates within the plaque cells, then used a random integer generator to select pixels by their list index. We used the randomly selected pixel coordinates as simulated spots and counted taxon-spot spatial associations as described above. This was repeated for 1000 simulations and we calculated the mean and standard deviation for the count values for each taxon. We then calculated the z-score for the count values: z=(count - mean) / standard deviation. P-values were calculated by counting the fraction of simulations with greater values than the observed value.

Statistics. Python v3.8.5 was used to generate statistics. Box plots consist of a bottom line representing the lower quartile (Q1), a line inside the box representing the median (Q2), a top line representing the upper quartile (Q3), an upper whisker extending from the top of the box indicating the maximum value within 1.5 times the interquartile range (IQR) above Q3, and a lower whisker extending from the bottom of the box indicating the minimum value within 1.5 times the IQR below Q1. Monte Carlo methods with 100 or 1000 simulations were used in two-sided tests to evaluate null hypotheses of random distribution of spots.

The specific implementation of code to generate figures presented here is available on GitHub at https://github.com/benjamingrodner/ hipr_mge_fish (v1.0.0, https://zenodo.org/doi/10.5281/zenodo.11085744). The generalized pipeline for segmentation is available at https:// github.com/benjamingrodner/pipeline_segmentation (v1.0.0, https://doi.org/10.5281/zenodo.11085837), while the generalized implementation of probe design is available at  https://github.com/benjamingrodner/FISH_split_probe_design (v1.0.0, https://

doi.org/10.5281/zenodo.11085839).

For manuscripts utilizing custom algorithms or software that are central to the research but not yet described in published literature, software must be made available to editors and reviewers. We strongly encourage code deposition in a community repository (e.g. GitHub). See the Nature Portfolio guidelines for submitting code & software for further information.

# Data

Policy information about availability of data

All manuscripts must include a data availability statement. This statement should provide the following information, where applicable:
- Accession codes, unique identifiers, or web links for publicly available datasets
- A description of any restrictions on data availability
- For clinical datasets or third party data, please ensure that the statement adheres to our policy

Illumina and PacBIO sequencing data are available at the NCBI Sequence Read Archive (SRA) with accession number PRJNA981198. Microscopy data have been deposited to Zenodo at https://doi.org/10.5281/zenodo.8015720 (Fig. 1b, Fig. S1a,b), https://doi.org/10.5281/zenodo.8015754 (Fig. 1c, Extended Data Fig. 1c,d, including count tables), https://doi.org/10.5281/zenodo.8015832 (Fig. 2, Extended Data Fig. 2, 3), https://zenodo.org/doi/10.5281/zenodo.11039333 (Fig. 3, Extended Data Fig. 4, including plasmid assembly with Illumina and Nanopore reads), and https://zenodo.org/doi/10.5281/zenodo.11039443 (Fig. 4, Extended Data Fig. 5, including plasmid assembly with Illumina and Nanopore reads).

GTDB release 207 is available at https://data.gtdb.ecogenomic.org/releases/release207/, CARD v3.1.0 is available at https://card.mcmaster.ca/download, PLSDB v.2023_11_03 is available at https://ccb-microbe.cs.uni-saarland.de/plsdb/plasmids/download/, RefSeq release 220 is available at https://ftp.ncbi.nlm.nih.gov/refseq/release/release-catalog/archive/, checkM database v2015-01-16 is available at https://zenodo.org/doi/10.5281/zenodo.7401544, geNomad database v1.7 is available at https://doi.org/10.5281/zenodo.10594875, and the Bakta database v5.0 is at https://doi.org/10.5281/zenodo.7669534. For VIBRANT, Pfam v32.0 is available https://ftp.ebi.ac.uk/pub/databases/Pfam/releases/Pfam32.0/, VOG v94 is available at https://fileshare.lisc.univie.ac.at/vog/vog94/, and KEGG v2019-03-20 is available at ftp://ftp.genome.jp/pub/db/kofam/archives/2019-03-20/.

# Research involving human participants, their data, or biological material

Policy information about studies with human participants or human data. See also policy information about sex, gender (identity/presentation), and sexual orientation and race, ethnicity and racism.

| Reporting on sex and gender | Sex and gender were not relevant in study design. |
|---|---|
| Reporting on race, ethnicity, or other socially relevant groupings | Socially relevant groupings were not relevant in study design. |
| Population characteristics | No covariate relevant population characteristics were considered in study design. |
| Recruitment | Two healthy volunteers donated samples were recruited Cornell. At Harvard School of Dental Medicine, one donor was recruited from patients in the advanced graduate periodontal department during their initial examination and/or dental hygiene therapy visit with inclusion criteria of age greater than 18 years and diagnosed periodontitis and exclusion criteria of greater than 20 cigarettes a day, antibiotic use within the last 8 weeks, and systemic condition requiring antibiotic prophylaxis. |
| Ethics oversight | The protocol for volunteer sample collection was approved by the Cornell Institutional Review Board (IRB) #2102010112. At Harvard School of Dental Medicine, IRB approval (IRB21-0662) was obtained for collection of patient specimens in the advanced graduate periodontal department. |

Note that full information on the approval of the study protocol must also be provided in the manuscript.

# Field-specific reporting

Please select the one below that is the best fit for your research. If you are not sure, read the appropriate sections before making your selection.

☒ Life sciences    ☐ Behavioural & social sciences    ☐ Ecological, evolutionary & environmental sciences

For a reference copy of the document with all sections, see nature.com/documents/nr-reporting-summary-flat.pdf

# Life sciences study design

All studies must disclose on these points even when the disclosure is negative.

| Sample size | No statistical methods were used to pre-determine sample sizes but our sample sizes are similar to those reported in previous publications. For cultured cell experiments, the number of cells measured was routinely in the thousands. Sample size was chosen based on fields of view, where each condition was measured with three tile scans composed of four fields of view each thus measuring thousands of cells. For oral plaque experiments, the target genes were unique to each volunteer, so multiple samples were not possible for a given target gene. Sample size was chosen based on fields of view, where each sample was measured with at least three tile scans composed of at least 9 fields of view each thus measuring thousands of cells. |
|---|---|

| | |
|---|---|
| Data exclusions | No data were excluded from the analysis. |
| Replication | All attempts at replication were successful. Multiple fields of view were collected for each sample. For oral plaque experiments, generally two technical replicates were performed. For cultured cell imaging, two replicates were performed. |
| Randomization | For technical controls, samples of cultured cells and plaque were allocated randomly. |
| Blinding | Blinding was not applicable in the methods development studies using cultured cells since imaging and image analysis settings were quantitatively standardized and replicated for negative and positive controls. Blinding was not needed for the descriptive and exploratory studies using plaque samples. No specific hypothesis was tested. |

# Reporting for specific materials, systems and methods

We require information from authors about some types of materials, experimental systems and methods used in many studies. Here, indicate whether each material, system or method listed is relevant to your study. If you are not sure if a list item applies to your research, read the appropriate section before selecting a response.

## Materials & experimental systems

| n/a | Involved in the study |
|---|---|
| ☒ | Antibodies |
| ☒ | Eukaryotic cell lines |
| ☒ | Palaeontology and archaeology |
| ☒ | Animals and other organisms |
| ☒ | Clinical data |
| ☒ | Dual use research of concern |
| ☒ | Plants |

## Methods

| n/a | Involved in the study |
|---|---|
| ☒ | ChIP-seq |
| ☒ | Flow cytometry |
| ☒ | MRI-based neuroimaging |

