## [Peer Review File · Nature Microbiology]

Peer Review Information

Journal: Nature Microbiology

Manuscript Title: Spatial Mapping of Mobile Genetic Elements and their Bacterial Hosts in Complex Microbiomes

Corresponding author name(s): Dr Iwijn De Vlaminc

Reviewer Comments & Decisions:

Decision Letter, initial version:

Message 2nd August 2023

:

Dear Dr De Vlaminck,

Thank you for your patience while your manuscript "Spatial Mapping of Mobile Genetic Elements and their Cognate Hosts in Complex Microbiomes" was under peer-review at Nature Microbiology. It has now been seen by 3 referees, whose expertise and comments you will find at the end of this email. Although they find your work of some potential interest, they have raised a number of concerns that will need to be addressed before we can consider publication of the work in Nature Microbiology.

In particular, you will see that referee #1 found it hard to identify single cells from some analyses and requests more insight into how MGE levels per cell were quantified, as well as a need to either check taxonomic identification or discuss discrepancies between the morphologies of certain taxa identified and what is generally described for these organisms. Referee #2 asks for more insight into sequencing depths, how MGEs were identified as MGEs and how plasmid carriage was demonstrated for certain genes probed. They also requested that this method be compared to existing methodologies while Referee #3 asked for a discussion of the potential limitations of the method. Several referees also questioned whether horizontal or vertical gene transfer is occurring and contributing to the MGE spots visualised. We feel that these are important points which should be addressed in a revised manuscript, alongside the rest of the referees' points which are clear and should be straightforward to address

Should further experimental data allow you to address these criticisms, we would be happy to look at a revised manuscript.

We strongly support public availability of data. Please place the data used in your paper into a public data repository, if one exists, or alternatively, present the data as Source Data or Supplementary Information. If data can only be shared on request, please explain why in your Data Availability Statement, and also in the correspondence with your editor. For some data types, deposition in a public repository is mandatory - more information on our data deposition policies and available repositories can be found at

2<https://www.nature.com/nature-research/editorial-policies/reporting-standards#availability-of-data>.

Please include a data availability statement as a separate section after Methods but before references, under the heading "Data Availability". This section should inform readers about the availability of the data used to support the conclusions of your study. This information includes accession codes to public repositories (data banks for protein, DNA or RNA sequences, microarray, proteomics data etc...), references to source data published alongside the paper, unique identifiers such as URLs to data repository entries, or data set DOIs, and any other statement about data availability. At a minimum, you should include the following statement: "The data that support the findings of this study are available from the corresponding author upon request", mentioning any restrictions on availability. If DOIs are provided, we also strongly encourage including these in the Reference list (authors, title, publisher (repository name), identifier, year). For more guidance on how to write this section please see:

<http://www.nature.com/authors/policies/data/data-availability-statements-data-citations.pdf>

* If you have not done so already we suggest that you begin to revise your manuscript so that it conforms to our Article format instructions at <http://www.nature.com/nmicrobiol/info/final-submission>. Refer also to any guidelines provided in this letter.

When submitting the revised version of your manuscript, please pay close attention to our [href="https://www.nature.com/nature-portfolio/editorial-policies/image-integrity">Digital Image Integrity Guidelines](https://www.nature.com/nature-portfolio/editorial-policies/image-integrity). and to the following points below:

3Please use the link below to submit a revised paper:

Note: This url links to your confidential homepage and associated information about manuscripts you may have submitted or be reviewing for us. If you wish to forward this e-mail to co-authors, please delete this link to your homepage first.

Nature Microbiology is committed to improving transparency in authorship. As part of our efforts in this direction, we are now requesting that all authors identified as 'corresponding author' on published papers create and link their Open Researcher and Contributor Identifier (ORCID) with their account on the Manuscript Tracking System (MTS), prior to acceptance. This applies to primary research papers only. ORCID helps the scientific community achieve unambiguous attribution of all scholarly contributions. You can create and link your ORCID from the home page of the MTS by clicking on 'Modify my Springer Nature account'. For more information please visit www.springernature.com/orcid.

If you wish to submit a suitably revised manuscript we would hope to receive it within 6 months. If you cannot send it within this time, please let us know. We will be happy to consider your revision, even if a similar study has been accepted for publication at Nature Microbiology or published elsewhere (up to a maximum of 6 months).

Reviewer Expertise:

Referee #1: spatial organisation of microbial communities
Referee #2: synthetic/systems biology, microbiomes, spatiotemporal metagenomics
Referee #3: AMR, environmental and gut microbiome resistome

Reviewer Comments:

Reviewer #1 (Remarks to the Author):

Review: Spatial Mapping of Mobile Genetic Elements and their Cognate Hosts in Complex Microbiomes

This is an important paper, jammed with information, well-written, well-documented and an excellent fit for Nature Microbiology. It addresses a major gap in understanding the biology of mobile genetic elements (MGEs)—namely, the lack of placing them in a spatial context. It achieves its goal by developing and applying imaging technology that can detect simultaneously the target genetic element and the taxonomic identities of the members of a

4microbiome in which it occurs.

The first section of the paper deals with the imaging methodology for detecting MGEs with convincing signal to noise ratios. This is introduced with the non-coding strand of GFP as target. The development of the technology through 6 different levels will be welcomed by researchers and is overall well-presented, but a reader might be confused on a few minor points. The legend explains for panel bii that magenta is the transformed plasmid but does not explicitly state what the teal color represents—is it the GFP expression or the *E. coli* 16S rRNA? In panel biii, spot fraction is plotted vs signal to noise ratio. But one has to dig through methods and guess how that ratio is calculated. Some brief explanatory text in the legend together with ref to methods would be useful. Visualizing T4 phage infection is beautiful and more powerfully convincing than the GFP results.

The authors applied their approach to complex microbiomes by detecting MGEs in oral plaque biofilms. They cleverly used metagenomic analysis to identify an antibiotic resistance gene located on a plasmid present in one subject but not in another, the negative subject serving as a control. They also probed for a T7-like phage and a gene, *termL*, present in a prevalent prophage. In addition, they attempted to visualize genes integrated on bacteria chromosomes. The results presented show target elements as micron scale clumps—which is a significant advance in itself—but does not clearly show individual bacterial cells. For this reason, I was puzzled by the bar diagram in Fig 2c denoting #spots/#cells. How were the cells counted? I cannot see them in Fig 2C. The authors should clarify how they did this quantification and refer to specific sections of methods as appropriate.

The authors then progress to combine MGE mapping with taxonomic identification. This is a major achievement. They associate MGEs such as *termL* with specific bacteria such as *Veillonella* by a proximity statistical association test. The test associating spots with cells is convincing but the bacterial cells themselves are not visualized crisply. The identification of specific taxa in Fig 3c is puzzling. The blue taxon identified as *Veillonella* appears filamentous whereas the green taxa identified as *Corynebacterium* appear coccoid. This is puzzling because *Corynebacteria* are known to be filamentous; not so for *Veillonella*. The authors should re-check their taxon identification and, if confirmed, say why they think it does not conform to established morphology.

In the Discussion, the authors “propose” that the localization of MGEs to spots/clusters represent short range MGE exchange or maintenance of MGEs through replication of host taxa. What other options are there? I find the language a bit odd. One would expect that clonal growth of a bacterium would propagate its MGEs and therefore make a spot. Therefore, spots per se do not distinguish between clonal growth and short range exchange. I suggest a more straightforward conclusion.

Overall this is a superb study and will likely be a widely read publication.

Reviewer #2 (Remarks to the Author):

Grodner et al describes an imaging approach using single molecule DNA FISH with multiplex rRNA fish to simultaneously visualize MGEs and host cells. The authors first test this approach for looking at phage infection and then track AMR plasmids in human oral community. Specific MGE and host co-localizations were identified.

Overall, the method is an interesting combination of HIPR-FISH that the authors previously

5developed and an improved protocol using split-HCR with gel embedding and clearing. There are some technical advances that seem to improve performance of the method. The biological insights generated in the study is rather limited for what the technique promises to be able to achieve.

Some comments:

1. How are the authors sure that the gene probed are indeed derived from plasmids?
2. How deep does one need to sequence the metagenome to identify the *mefE* AMR gene is originating from a plasmid or to identify the T7 like prophage? This seems like a very limiting step of the study and should be articulated more clearly.
3. There is little treatment for assessing whether the MGEs identified are being spread vertically or horizontally and whether these are truly MGEs.
4. The authors state that: "The plasmid's transfer from *Veillonella* to Pasteurellaceae or *Prevotella* seems unlikely given the significant phylogenetic distance between them, as HGT usually occurs between closely related species." This is rather speculative as past work has shown that distant HGT is possible and could be prevalent in some microbiome context.
5. Beyond some host-taxon and AMR gene associations, there is little analysis for overall distributions of the AMR genes and spatial arrangement.
6. Better highlight and compare how this method compares with other methods in spatial genomics using FISH and similar methods.

Reviewer #3 (Remarks to the Author):

This manuscript provides a major step forward in the development of tools to link mobile genetic elements to their microbial hosts in microbiomes. The authors use state-of-the-art techniques, combining shotgun metagenomics, single-molecule fluorescent in-situ hybridization (FISH) and multiplexed ribosomal RNA FISH to visualise mobile genetic elements (plasmids and phages) and their microbial hosts. In my opinion, this manuscript describes important work that is performed to high standards. Detailed suggestions for further improvements and corrections are outlined below. In addition, I suggest the authors add a short paragraph in the discussion to highlight limitations of the technique. These could include the relatively limited 'taxonomic resolution' of FISH-approaches, the need for highly specialised microscopy infrastructure (this is not a high-throughput technique) and questions around the reproducibility of this technique outside the niche (oral plaque biofilms of two volunteers) studied here. I also note that the oral plaque biofilms have been stored in 50% ethanol upon collection: could this step lead to lysis of the cells (losing potential MGE-host links)?

Fig. 1, panel a; l. 544; legend Fig S1; Fig S3, panel a: write 16S not 16s.

l. 25: the term 'specialized niches' might be construed as referring to a presumptive active process that creates these micro-environments to retain MGEs in bacterial populations. There is no evidence provided in this manuscript to support this statement and I would suggest it is rephrased or removed.

l. 27-28: the link to phage therapy does not follow logically from the rest of the abstract and should thus be removed.

6I. 42: a reference should probably be added to a paper that highlights the difficulties of linking MGEs to their microbial hosts in microbiomes.

I. 131: more information on *mefE* should be provided (e.g. identify to the closest hit in RefSeq and %identity to the gene identified here in a Supplementary Table). It is my understanding that *mefE* is mostly found in Bacillota/Firmicutes, class Bacilli, so the link to *Veillonella* is somewhat surprising, but not entirely implausible. Has *mefE* been found in *Veillonella* before?

I. 167: more information on *patA*, *patB*, and *adeF* should be provided (e.g. identify to the closest hit in RefSeq and %identity to the gene identified here in a Supplementary Table)

I. 204: couldn't should be written as could not.

I. 224. Add a reference to the statement that HGT usually occurs between closely related species.

I. 256: I believe 'this' is better replaced by 'the'.

I. 340: add reference to 'Method d from DNA-FISH protocols'

Table S1: text seems to be cut off from the left side of the table.

This peer review report was contributed by Prof Willem van Schaik, University of Birmingham, United Kingdom.

Author Rebuttal to Initial comments

Point-by-point response to the referees' comments (Our response in blue font)

Reviewer #1 (Remarks to the Author):

This is an important paper, jammed with information, well-written, well-documented and an excellent fit for Nature Microbiology. It addresses a major gap in understanding the biology of mobile genetic elements (MGEs)—namely, the lack of placing them in a spatial context. It

7achieves its goal by developing and applying imaging technology that can detect simultaneously the target genetic element and the taxonomic identities of the members of a microbiome in which it occurs.

Thank you for your appreciation of our work. In the revised manuscript, we have addressed all your comments and suggestions. We also include data from new experiments, including new MGE-FISH data for an additional AMR plasmid, corroborating its association with a previously identified host, and data from new experiments visualizing a highly abundant plasmid in a plaque biofilm from a patient with stage 3 periodontitis, using our method to determine its cognate bacterial host.

The first section of the paper deals with the imaging methodology for detecting MGEs with convincing signal to noise ratios. This is introduced with the non-coding strand of GFP as target. The development of the technology through 6 different levels will be welcomed by researchers and is overall well-presented, but a reader might be confused on a few minor points. The legend explains for panel bii that magenta is the transformed plasmid but does not explicitly state what the teal color represents—is it the GFP expression or the *E. coli* 16S rRNA? In panel biii, spot fraction is plotted vs signal to noise ratio. But one has to dig through methods and guess how that ratio is calculated. Some brief explanatory text in the legend together with ref to methods would be useful. Visualizing T4 phage infection is beautiful and more powerfully convincing than the GFP results.

Thank you for your comments on the presentation of this figure. The teal color is from 16S rRNA staining and we clarified this in the legend and caption of figure 1. We also clarified the calculation of signal to noise ratio in the caption of figure 1.

The authors applied their approach to complex microbiomes by detecting MGEs in oral plaque biofilms. They cleverly used metagenomic analysis to identify an antibiotic resistance gene located on a plasmid present in one subject but not in another, the negative subject serving as a control. They also probed for a T7-like phage and a gene, *termL*, present in a prevalent prophage. In addition, they attempted to visualize genes integrated on bacteria chromosomes. The results presented show target elements as micron scale clumps—which is a significant advance in itself—but does not clearly show individual bacterial cells. For this reason, I was puzzled by the bar diagram in Fig 2c denoting #spots/#cells. How were the cells counted? I cannot see them in Fig 2C. The authors should clarify how they did this quantification and refer to specific sections of methods as appropriate.

Thank you for pointing this out. In the new version of the manuscript, we add information on how we quantified the #spots per #cells, see the caption of Figure 2. Further, we include an example of the segmentation and spot counting in Supplementary Figure S2a.

The authors then progress to combine MGE mapping with taxonomic identification. This is a major achievement. They associate MGEs such as *termL* with specific bacteria such as *Veillonella* by a proximity statistical association test. The test associating spots with cells is convincing but the bacterial cells themselves are not visualized crisply.

We have updated the visualization of bacterial cells in Figure 3c.

The identification of specific taxa in Fig 3c is puzzling. The blue taxon identified as *Veillonella* appears filamentous whereas the green taxa identified as *Corynebacterium* appear coccoid. This is puzzling because *Corynebacteria* are known to be filamentous; not so for *Veillonella*. The authors should re-check their taxon identification and, if confirmed, say why they think it does not conform to established morphology.

We re-checked the spectra of the filamentous cells and found that they have the correct spectrum. We have two potential explanations for this: these cells are not *Veillonella* and thus both the *Veillonella* 16s rRNA FISH probes and the *termL* probes bind off-target to these filamentous cells, or, potentially, these cells are *Veillonella* in which phage are actively replicating and the filamentous morphology is a stress response to the infection as we observed in *E. coli* (e.g. Fig. 1c, 20min MOI 0.01 and 0.1). Others have also demonstrated this response in *E. coli*, though for other stressors (<https://doi.org/10.1111/mmi.15016>). We have included a discussion of this observation in the text. We also re-checked the fluorescent spectra of the cells classified as *Corynebacterium* and found that they were mislabeled; the fluorescent spectra indicate that the cells are *Lautropia*. Thank you for pointing this out.

In the Discussion, the authors “propose” that the localization of MGEs to spots/clusters represent short range MGE exchange or maintenance of MGEs through replication of host taxa. What other options are there? I find the language a bit odd. One would expect that clonal growth of a bacterium would propagate its MGEs and therefore make a spot. Therefore, spots per se do not distinguish between clonal growth and short range exchange. I suggest a more straightforward conclusion.

We agree and have updated the discussion section accordingly.

Overall this is a superb study and will likely be a widely read publication.

Thank you again for your thoughts and comments!

Reviewer #2 (Remarks to the Author):

Grodner et al describes an imaging approach using single molecule DNA FISH with multiplex rRNA fish to simultaneously visualize MGEs and host cells. The authors first test this approach for looking at phage infection and then track AMR plasmids in human oral community. Specific MGE and host co-localizations were identified.

Overall, the method is an interesting combination of HIPR-FISH that the authors previously developed and an improved protocol using split-HCR with gel embedding and clearing. There are some technical advances that seem to improve performance of the method. The biological

insights generated in the study is rather limited for what the technique promises to be able to achieve.

Thank you for your comments and questions, which have helped us to further improve our manuscript and analysis. The most important changes beyond improvements in the text and presentation of the data include *i)* the addition of long-read nanopore sequencing data to facilitate complete assembly of targeted AMR plasmids; *ii)* Inclusion of new MGE-FISH data for an additional AMR plasmid, corroborating its association with a previously identified host; *iii)* new analyses of the spatial distribution of plasmids within plaque biofilms; and *iv)* data from new experiments visualizing a highly abundant plasmid in a plaque biofilm from a patient with stage 3 periodontitis, using our method to determine its cognate bacterial host.

Some comments:

1. How are the authors sure that the gene probed are indeed derived from plasmids?

Prompted by your question we inspected the contig containing *mefE* more carefully and are now no longer convinced that this contig is derived from a plasmid. The contig is too short and does not contain genes that would make it mobilizable. In our initial analyses, we used SCAPP to identify circular paths in the assembly graphs, but solely relying on SCAPP to predict plasmid nature was a weak assumption. Dissatisfied with this, we employed long read nanopore sequencing on our volunteer samples to improve contig assembly and identified several contigs that contained predicted plasmid mobilization genes and that all aligned with near perfect homology to a known plasmid in a *Prevotella* strain. We then used our imaging approach to map the plasmid and found that it colocalized with *Prevotella*. Wishing to extend our data beyond validation experiments, we obtained a clinical plaque sample from a patient with periodontitis and identified a predicted plasmid with a complete circular sequence, multiple plasmid mobilization genes, and a plasmid-like GC skew pattern. We employed our imaging approach to identify *Streptococcus* as the host.

2. How deep does one need to sequence the metagenome to identify the *mefE* AMR gene is originating from a plasmid or to identify the T7 like prophage? This seems like a very limiting step of the study and should be articulated more clearly.

This is an important point, and we have added a sentence to the Discussion section addressing the sequencing required to identify MGEs for this method.

3. There is little treatment for assessing whether the MGEs identified are being spread vertically or horizontally and whether these are truly MGEs.

This follows on remarks from Reviewer #1, who noted that clusters of plasmids or prophage could arise from clonal growth or short-range horizontal transfer. In the new version of the manuscript,

11we acknowledge that our data cannot distinguish between these two possibilities.

4. The authors state that: “The plasmid's transfer from Veillonella to Pasteurellaceae or Prevotella seems unlikely given the significant phylogenetic distance between them, as HGT usually occurs between closely related species.” This is rather speculative as past work has shown that distant HGT is possible and could be prevalent in some microbiome context.

This is a good point, and we have updated the text in the “Combined taxonomic mapping and MGE mapping” section and removed speculation on the likelihood of distant HGT.

5. Beyond some host-taxon and AMR gene associations, there is little analysis for overall distributions of the AMR genes and spatial arrangement.

To address this comment, we have added a supplementary figure S4 which analyzes the distribution of a putative metallo- β -lactamase gene.

6. Better highlight and compare how this method compares with other methods in spatial genomics using FISH and similar methods.

To address this comment, we have added a paragraph to the Discussion comparing this method to other methods addressing a similar problem.

Reviewer #3 (Remarks to the Author):

This manuscript provides a major step forward in the development of tools to link mobile genetic elements to their microbial hosts in microbiomes. The authors use state-of-the-art techniques, combining shotgun metagenomics, single-molecule fluorescent in-situ hybridization (FISH) and multiplexed ribosomal RNA FISH to visualise mobile genetic elements (plasmids and phages) and their microbial hosts. In my opinion, this manuscript describes important work that is performed to high standards.

Thank you for your comments on the strong points of our work as well as your suggestions for further improvements. We have followed all your suggestions, and we also include data from extensive new experiments. We include a new example of an AMR plasmid with a known host in RefSeq, which we used to further validate our combined plasmid and taxon visualization method. In another new experiment, we used our method to identify the cognate bacterial host of a plasmid which had no homologs in RefSeq, but is circular, has appropriate plasmid mobilization genes, and exhibits characteristic GC skew.

Detailed suggestions for further improvements and corrections are outlined below. In addition, I suggest the authors add a short paragraph in the discussion to highlight limitations of the technique. These could include the relatively limited 'taxonomic resolution' of FISH-approaches, the need for highly specialized microscopy infrastructure (this is not a high-throughput technique) and questions around the reproducibility of this technique outside the niche (oral plaque biofilms of two volunteers) studied here.

Thank you for the suggestion, we discuss limitations of the technique in the Discussion section of the revised manuscript.

I also note that the oral plaque biofilms have been stored in 50% ethanol upon collection: could this step lead to lysis of the cells (losing potential MGE-host links)?

While this is certainly a possibility, we believe that it has minimal impact in these samples. Based on our experiments, our assumption is that there is limited loss of plasmids or phage through membrane disruption in ethanol storage because diffusion is limited by the cell wall and dehydration-based fixation of the biofilm.

Fig. 1, panel a; l. 544; legend Fig S1; Fig S3, panel a: write 16S not 16s.

We updated all instances of '16s' to '16S'.

I. 25: the term 'specialized niches' might be construed as referring to a presumptive active process that creates these micro-environments to retain MGEs in bacterial populations. There is no evidence provided in this manuscript to support this statement and I would suggest it is rephrased or removed.

Thank you for pointing this out, we have removed all instances of this term and rephrased to avoid this statement.

I. 27-28: the link to phage therapy does not follow logically from the rest of the abstract and should thus be removed.

We have changed the abstract to address this point.

I. 42: a reference should probably be added to a paper that highlights the difficulties of linking MGEs to their microbial hosts in microbiomes.

We have added a reference to Brito IL, 2021 (<https://doi.org/10.1038/s41579-021-00534-7>).

I. 131: more information on *mefE* should be provided (e.g. identify to the closest hit in RefSeq and %identity to the gene identified here in a Supplementary Table). It is my understanding that *mefE* is mostly found in Bacillota/Firmicutes, class Bacilli, so the link to *Veillonella* is somewhat surprising, but not entirely implausible. Has *mefE* been found in *Veillonella* before?

We have added Refseq alignment information for all target genes to the Supplementary Table 10, and we have added references to it in the text. Following on our response to questions from Reviewer #2 we have removed claims about *mefE* host association due to lack of evidence of occurrence on a mobile plasmid.

I. 167: more information on *patA*, *patB*, and *adeF* should be provided (e.g. identify to the closest hit in RefSeq and %identity to the gene identified here in a Supplementary Table)

As per the previous suggestion, we have added Refseq alignment information for these genes to the Supplementary Table 10.

I. 204: couldn't should be written as could not.

Thank you for pointing this out, we changed the "couldn't" to "could not".

I. 224. Add a reference to the statement that HGT usually occurs between closely related species.

This follows on remarks from Reviewer #2, who pointed out that previous work has shown distant HGT is possible and could be prevalent in certain contexts. To address this, we have removed speculation as to the likelihood of HGT between distantly related species.

I. 256: I believe 'this' is better replaced by 'the'.
We have followed this suggestion.

I. 340: add reference to 'Method d from DNA-FISH protocols'

We have followed this suggestion.

Table S1: text seems to be cut off from the left side of the table.

Thank you for pointing this out, we have corrected the Supplementary Tables.

Decision Letter, first revision:

Message: Our ref: NMICROBIOL-23061400A

25th April 2024

Dear Dr. De Vlamincx,

Thank you for your patience as we've prepared the guidelines for final submission of your Nature Microbiology manuscript, "Spatial Mapping of Mobile Genetic Elements and their Cognate Hosts in Complex Microbiomes" (NMICROBIOL-23061400A). Please carefully follow the step-by-step instructions provided in the attached file, and add a response in each row of the table to indicate the changes that you have made. Ensuring that each point is addressed will help to ensure that your revised manuscript can be swiftly handed over to our production team.

If you have not done so already, please alert us to any related manuscripts from your

16group that are under consideration or in press at other journals, or are being written up for submission to other journals (see: <https://www.nature.com/nature-research/editorial-policies/plagiarism#policy-on-duplicate-publication> for details).

In recognition of the time and expertise our reviewers provide to Nature Microbiology's editorial process, we would like to formally acknowledge their contribution to the external peer review of your manuscript entitled "Spatial Mapping of Mobile Genetic Elements and their Cognate Hosts in Complex Microbiomes". For those reviewers who give their assent, we will be publishing their names alongside the published article.

Nature Microbiology offers a Transparent Peer Review option for new original research manuscripts submitted after December 1st, 2019. As part of this initiative, we encourage our authors to support increased transparency into the peer review process by agreeing to have the reviewer comments, author rebuttal letters, and editorial decision letters published as a Supplementary item. When you submit your final files please clearly state in your cover letter whether or not you would like to participate in this initiative. Please note that failure to state your preference will result in delays in accepting your manuscript for publication.

Cover suggestions

COVER ARTWORK: We welcome submissions of artwork for consideration for our cover. For more information, please see our guide for cover artwork.

Nature Microbiology has now transitioned to a unified Rights Collection system which will allow our Author Services team to quickly and easily collect the rights and permissions required to publish your work. Approximately 10 days after your paper is formally accepted, you will receive an email in providing you with a link to complete the grant of rights. If your paper is eligible for Open Access, our Author Services team will also be in touch regarding any additional information that may be required to arrange payment for your article.

Please note that *Nature Microbiology* is a Transformative Journal (TJ). Authors may publish their research with us through the traditional subscription access route or make their paper immediately open access through payment of an article-processing charge (APC). Authors will not be required to make a final decision about access to their article until it has been accepted. Find out more about Transformative Journals

Authors may need to take specific actions to achieve compliance with funder and institutional open access mandates. If your research is supported by a funder that requires immediate open access (e.g. according to Plan S principles) then you should select the gold OA route, and we will direct you to the compliant route where possible. For authors selecting the subscription publication route, the journal's standard licensing terms will need to be accepted, including self-archiving policies. Those licensing terms will

2supersede any other terms that the author or any third party may assert apply to any version of the manuscript.

Best regards,

Reviewer #1:

Remarks to the Author:

The authors have responded satisfactorily to all points raised in the reviews. An excellent paper improved even further.

Reviewer #2:

Remarks to the Author:

The reviewer appreciates the author's revised manuscript and efforts to improve the analysis and addition of new data. The revised manuscript addresses the reviewer's outstanding concerns. It's a nice paper that describes an advanced method to map out spatial organization of mobile elements in a microbial community.

Reviewer #3:

Remarks to the Author:

This manuscript has further improved upon revision. I only have very minor comments

L. 164: bacteria should be bacterium

L. 225: what is meant with 'These cells have the correct spectrum for Veillonella'? I guess this refers to the fluorescent Veillonella probe binding to these cells?

Author Rebuttal, first revision:

3Point-by-point response to the referees' comments

(Our response in blue font)

Reviewer #1: Remarks to the Author:

The authors have responded satisfactorily to all points raised in the reviews. An excellent paper improved even further.

Thank you for your appreciation of our work.

Reviewer #2: Remarks to the Author:

The reviewer appreciates the author's revised manuscript and efforts to improve the analysis and addition of new data. The revised manuscript addresses the reviewer's outstanding concerns. It's a nice paper that describes an advanced method to map out spatial organization of mobile elements in a microbial community.

Thank you for your appreciation of our work.

Reviewer #3: Remarks to the Author:

This manuscript has further improved upon revision. I only have very minor comments

Thank you, in the revised manuscript, we have addressed all your comments.

L. 164: bacteria should be bacterium

We have made this change.

L. 225: what is meant with 'These cells have the correct spectrum for *Veillonella*'? I guess this refers to the fluorescent *Veillonella* probe binding to these cells?

We have altered the sentence to clarify that these cells are stained with the fluorescent barcode we assigned to *Veillonella*.

Final Decision Letter:

Message: 17th May 2024

Dear Dr De Vlaminck,

I am pleased to accept your Article "Spatial Mapping of Mobile Genetic Elements and their Bacterial Hosts in Complex Microbiomes" for publication in Nature Microbiology. Thank you for having chosen to submit your work to us and many congratulations.

You may wish to make your media relations office aware of your accepted publication, in case they consider it appropriate to organize some internal or external publicity. Once your paper has been scheduled you will receive an email confirming the publication details. This is normally 3-4 working days in advance of publication. If you need additional notice of the date and time of publication, please let the production team know when you receive the proof of your article to ensure there is sufficient time to coordinate. Further information on our embargo policies can be found here:

<https://www.nature.com/authors/policies/embargo.html>

After the grant of rights is completed, you will receive a link to your electronic proof via email with a request to make any corrections within 48 hours. If, when you receive your proof, you cannot meet this deadline, please inform us at

5rjsproduction@springernature.com immediately. You will not receive your proofs until the publishing agreement has been received through our system

Please note that *Nature Microbiology* is a Transformative Journal (TJ). Authors may publish their research with us through the traditional subscription access route or make their paper immediately open access through payment of an article-processing charge (APC). Authors will not be required to make a final decision about access to their article until it has been accepted. Find out more about Transformative Journals

You can now use a single sign-on for all your accounts, view the status of all your manuscript submissions and reviews, access usage statistics for your published articles and

download a record of your refereeing activity for the Nature journals.

With kind regards,